# Geometry-Aware Contrastive Learning for Few-Shot Automatic Modulation Recognition

Guanqun Zhao [1]   Yitong Liu [1]   Jiaxuan Fang [1]   Yufei Mao [1]   Hongwen Yang [1]

## Abstract

Standard Self-Supervised Learning (SSL) for Automatic Modulation Recognition (AMR) struggles with ineffective isotropic augmentations, spectral instability, and semantic drift. To address these challenges, we propose Dynamic-Consistency Contrastive Learning (DyCo-CL), a geometry-aware framework that couples Virtual Adversarial Augmentation (VAA) with a semantic consistency loss. We provide a theoretical analysis indicating that this strategy acts as an implicit spectral regularizer for the encoder, enabling stable manifold exploration. Complementing this, our Signal-Adaptive Swin Backbone with fixed-window attention improves structural stability by constraining attention locality, while a Hybrid Knowledge Fusion module anchors representations with physical priors. Experiments on RML benchmarks show that DyCo-CL achieves a 6.27% accuracy gain in 1-shot settings over prior methods.

## 1. Introduction

Automatic Modulation Recognition (AMR) serves as the cornerstone of cognitive radio, enabling dynamic spectrum access in emerging 6G networks. While Deep Learning (DL) has superseded traditional expert-based methods in terms of complexity and representational power (Zhou et al., 2020; Zhang et al., 2025), its efficacy remains contingent on massive labeled datasets, a luxury often unavailable in non-cooperative environments.

To mitigate data scarcity, Self-Supervised Learning (SSL) has emerged as the dominant paradigm. Researchers have adapted contrastive frameworks to RF signals (Davaslioglu et al., 2023; Chen et al., 2025), typically relying on stochas-tic data augmentations to learn invariant representations. To further boost robustness, recent works have begun to integrate physical priors or employ lightweight attention mechanisms (Kong et al., 2025; Ma et al., 2026; Deng et al., 2023).

However, these methods operate on the flawed assumption that standard augmentations and generic attention remain robust in high-dimensional spaces. By analyzing the signal manifold, we identify three critical geometric limitations that hinder current frameworks:

**The Geometric Dilemma of Augmentation.** High-dimensional concentration of measure (Walters, 2015) renders isotropic noise ineffective as perturbations tend to be orthogonal to decision boundary gradients. Meanwhile, unconstrained transformations often breach class margins, inducing semantic drift.

**Spectral Instability of Self-Attention.** Self-attention exhibits unbounded Lipschitz constants (Kim et al., 2021), resulting in sharp decision boundaries that are brittle to anisotropic perturbations. Existing hybrid architectures (Kong et al., 2025; Ma et al., 2026) prioritize efficiency but neglect this instability, leaving models vulnerable to distortion.

**Inefficacy of Static Fusion.** Prevalent methods rely on shallow concatenation (Deng et al., 2023), treating physical priors as static auxiliary inputs (Feng et al., 2025). This fails to leverage them as immutable semantic anchors, preventing the rectification of semantic drift during aggressive exploration, especially in 1-shot settings.

To address these challenges, we propose DyCo-CL, a semi-supervised framework synergizing optimization, architecture, and physical priors. Our contributions are:

- **Dynamic-Consistency Framework:** We propose a geometric optimization strategy that couples Virtual Adversarial Augmentation (VAA) with a semantic consistency constraint. This approach overcomes the *concentration of measure* where isotropic noise fails, and we theoretically characterize it as an implicit spectral regularizer to promote the geometric stability of the encoder.

[1] Beijing University of Posts and Telecommunications, Beijing, China. Correspondence to: Yitong Liu <liuyitong@bupt.edu.cn>.

*Proceedings of the 43rd International Conference on Machine Learning*, Seoul, South Korea. PMLR 306, 2026. Copyright 2026 by the author(s).

- **Signal-Adaptive Swin Backbone:** To complement the geometric regularization, we design a 1D Swin Transformer featuring a Deep Convolutional Stem and Fixed Window attention. This architecture addresses the spectral instability of standard Transformers, enabling the model to robustly capture transient signal primitives under adversarial perturbations.

- **Hierarchical Hybrid Knowledge Fusion:** To prevent semantic drift in data-scarce regimes, we bridge the semantic gap via a physics-aware fusion module. By dynamically calibrating deep representations with expert physical priors , we ensure that the manifold exploration remains physically grounded, essential for extreme 1-shot recognition.

## 2. Related Work

### 2.1. Contrastive Learning for Modulation Recognition

While early AMR methods relied on manual feature extraction (Dan et al., 2005; Hazza et al., 2013) or likelihood-based inference (Xu et al., 2010; Zheng & Lv, 2018), these approaches struggle with modeling complexity. Consequently, DL has become the dominant paradigm. For instance, (Zhou et al., 2020) proposed an LSTM-based method to effectively extract spatiotemporal features for modulation classification, while (Zhang et al., 2025) utilized ResNet to improve recognition accuracy in wireless communication systems. To further enhance model robustness, (Liang et al., 2025) introduced fuzzy regularization into the classification framework. SSL has recently emerged to address data scarcity, with frameworks like MoCo (Chen & Xie, 2021) and SimCLR (Chen et al., 2020) adapted to RF signals (Davaslioglu et al., 2023; Chen et al., 2025).

To further boost performance, recent studies have begun to explore advanced augmentation strategies. Notably, SSCL-AMC (Cai et al., 2025) introduces gradient-based adversarial augmentation to mine hard samples. However, these approaches operate primarily at the data level, lacking explicit spectral constraints to guarantee the geometric stability of the learned representation against perturbations.

### 2.2. Backbone Architectures: From CNNs to Transformers

Convolutional Neural Networks (CNNs) have long been the de facto standard for robust AMR, with architectures explicitly designed to handle multipath fading and channel impairments (Tekbıyık et al., 2020). Recently, Transformers have attracted attention for their sequence modeling capabilities (Chen et al., 2025; Ma et al., 2026). Despite their potential, standard Transformers are known to yield sharp decision boundaries due to the unbounded Lipschitz constant of the self-attention mechanism (Kim et al., 2021),

making them brittle to anisotropic noise. Although hybrid architectures (Kong et al., 2025; Ma et al., 2026) improve efficiency, they typically lack structural mechanisms to explicitly bound the spectral norm, leaving the model vulnerable to adversarial stress.

### 2.3. Physics-Aware Knowledge Fusion

Integrating expert knowledge is a proven strategy to enhance robustness. Existing methods typically employ a dual-stream approach (Feng et al., 2025; Liu et al., 2024), merging physical and deep features (Lu et al., 2025) via simple concatenation (Bai et al., 2024; Deng et al., 2023). While effective, these strategies treat physical priors as static auxiliary inputs (Sümen et al., 2022), failing to bridge the *semantic gap* required to dynamically calibrate deep features in extreme few-shot regimes.

## 3. Preliminaries

We formulate the AMR signal model and analyze the geometric landscape, identifying three bottlenecks: concentration of measure, semantic drift, and spectral instability that motivate the geometry-aware design of DyCo-CL.

### 3.1. Signal Model

We formulate AMR as a classification task for complex-valued radio signals. A received signal frame of length $L$ is represented as a real-valued tensor $\mathbf{x} \in \mathbb{R}^{2 \times L}$, consisting of In-Phase ($I$) and Quadrature ($Q$) components. The observed signal is modeled as:

$$\mathbf{x} = h(\mathbf{s}) + \mathbf{n}, \tag{1}$$

where $\mathbf{s}$ is the clean modulated signal , $h(\cdot)$ denotes channel impairments, and $\mathbf{n}$ represents additive noise. Our objective is to learn a mapping $f_\theta : \mathbb{R}^{2 \times L} \to \mathcal{Y}$ that predicts the modulation label $y \in \mathcal{Y}$ given the observation $\mathbf{x}$.

### 3.2. Problem Formulation: Concentration of Measure and Geometric Dilemma

Let the signal space be $\mathcal{X} \subset \mathbb{R}^D$ with $D = 2L$. From a geometric perspective, the core challenge lies in learning an encoder $f_\theta$ that maintains class separability while ensuring invariance to channel perturbations. Standard contrastive learning minimizes representation divergence over a distribution of transformations $\mathcal{T}$. We analyze the geometric limitations of this paradigm below.

#### 3.2.1. THE ORTHOGONALITY OF ISOTROPIC NOISE

Consider the set of isotropic augmentations $\mathcal{T}_{iso}$, typically implemented as additive white Gaussian noise: $t(\mathbf{x}) = \mathbf{x} + \mathbf{r}$, where $\mathbf{r} \sim \mathcal{N}(0, \sigma^2 \mathbf{I}_D)$. Let $\mathbf{v} = \nabla_{\mathbf{x}} \mathcal{L} / \|\nabla_{\mathbf{x}} \mathcal{L}\|$ be the unit

vector representing the *sensitive direction*. Here, $\mathcal{L}$ denotes the self-supervised contrastive loss ($\mathcal{L}_{\text{NCE}}$), implying that $\mathbf{v}$ is the direction that maximally disrupts the feature consistency between positive pairs. We decompose the perturbation $\mathbf{r}$ into sensitive and tangent components:

$$\mathbf{r} = z\mathbf{v} + \mathbf{r}_\perp, \quad \text{where } z = \langle \mathbf{r}, \mathbf{v} \rangle \sim \mathcal{N}(0, \sigma^2). \quad (2)$$

In high dimensions, the perturbation magnitude concentrates around its mean: $\|\mathbf{r}\| \approx \sigma\sqrt{D}$.

**Vanishing Sensitive Projection.** Applying the Gaussian tail inequality, the probability that the projection $z$ accounts for a fraction $\alpha$ of the total perturbation scale is bounded by:

$$P(|z| \geq \alpha\sigma\sqrt{D}) \leq 2\exp\left(-\frac{D\alpha^2}{2}\right). \quad (3)$$

This indicates that the effective perturbation along the gradient decays exponentially with $D$.

**Almost Sure Orthogonality.** Defining the alignment angle $\theta$ via $\cos\theta = z/\|\mathbf{r}\|$, the concentration of measure on $\mathbb{S}^{D-1}$ (Walters, 2015) implies that the probability mass concentrates on the equator $\mathbf{v}^\perp$. For any small angular tolerance $\delta > 0$, the probability of deviating from orthogonality is bounded by:

$$P(|\cos\theta| \geq \delta) \leq 2\exp\left(-\frac{D\delta^2}{2}\right). \quad (4)$$

For $D \approx 256$, numerical analysis confirms that significant alignment is statistically rare (see quantitative verification in Appendix A.1). While not strictly impossible, this implies $\mathbf{r} \perp \mathbf{v}$ with high probability. Consequently, optimizing over $\mathcal{T}_{iso}$ *inefficiently* reduces to minimizing: $\mathbb{E}_{\mathbf{r}\in\mathcal{T}_{iso}}[\mathcal{L}(f(\mathbf{x}), f(\mathbf{x}+\mathbf{r}_\perp))]$, enforcing invariance predominantly along "safe" tangent directions. This creates a Robustness Illusion: the model tolerates high-energy noise but remains brittle along the critical direction $\mathbf{v}$.

### 3.2.2. THE RISK OF SEMANTIC DRIFT

Geometric augmentations $\mathcal{T}_{geo}$, while anisotropic, are model-agnostic. Since physical signal boundaries are fixed, blind transformations often traverse the inter-class margins (e.g., rotating a QPSK symbol into an adjacent quadrant). We define this phenomenon as Semantic Drift: a conflict where the true physical label changes due to excessive perturbation, yet the contrastive loss forces the model to align the augmented view with the original anchor. This introduces noisy gradients that distort the learned manifold geometry.

### 3.2.3. THE SPECTRAL INSTABILITY OF TRANSFORMERS

Beyond augmentation, the encoder architecture itself poses a geometric risk. We characterize the geometric stability of $f_\theta$ via the local Lipschitz constant:

$$K_f(\mathbf{x}) \triangleq \sup_{\|\mathbf{r}\|\leq\epsilon} \frac{\|f_\theta(\mathbf{x}+\mathbf{r}) - f_\theta(\mathbf{x})\|}{\|\mathbf{r}\|}. \quad (5)$$

(Kim et al., 2021) proved that the Lipschitz constant of standard dot-product self-attention is *unbounded* with respect to sequence length $L$. This implies that naive Transformers inherently learn sharp decision boundaries, making them brittle to the anisotropic perturbations required to overcome the concentration of measure.

## 4. Methodology

### 4.1. Overview of DyCo-CL

We propose DyCo-CL, a geometry-aware framework tailored for few-shot AMR. As illustrated in Fig. 1, the system orchestrates three interdependent components: (1) a Dynamic-Consistency Framework utilizing VAA to overcome the concentration of measure; (2) a Signal-Adaptive Swin Backbone ensuring structural stability; and (3) Hierarchical Hybrid Knowledge Fusion, which anchors features with physical priors to counteract semantic drift. Together, they form a closed-loop system where geometric and physical constraints mutually reinforce robustness.

### 4.2. Dynamic-Consistency Framework

Built upon MoCov3, we address the failure of isotropic noise due to the *concentration of measure* (Sec. 3.2). We propose a *Dynamic-Consistency* strategy that couples VAA (to target sensitive directions) with a semantic alignment constraint (to ensure geometric stability).

#### 4.2.1. ASYMMETRIC AUGMENTATION STRATEGY

We employ an asymmetric design to construct robust positive pairs. One branch applies standard physical transformations to simulate channel impairments, while the other utilizes *VAA* to generate "hard positives".

**Virtual Adversarial Augmentation.** Unlike random noise which is almost surely orthogonal to the gradient, VAA actively seeks the perturbation direction that maximally alters the output distribution. We define the prediction probability $P(\cdot|\mathbf{x};\theta)$ as the softmax-normalized distribution of similarities between the query $\mathbf{q} = f_\theta(\mathbf{x})$ and the dictionary keys $\{\mathbf{k}_i\}$:

$$P(i|\mathbf{x};\theta) = \frac{\exp(\mathbf{q}\cdot\mathbf{k}_i/\tau)}{\sum_j \exp(\mathbf{q}\cdot\mathbf{k}_j/\tau)}, \quad (6)$$

where $\tau$ is the temperature parameter. Using this distribution, we quantify local sensitivity via the Kullback-Leibler (KL) divergence:

$$\mathcal{J}(\mathbf{r}) = \text{KL}\left[P(\cdot|\mathbf{x};\theta)\|P(\cdot|\mathbf{x}+\mathbf{r};\theta)\right]. \quad (7)$$

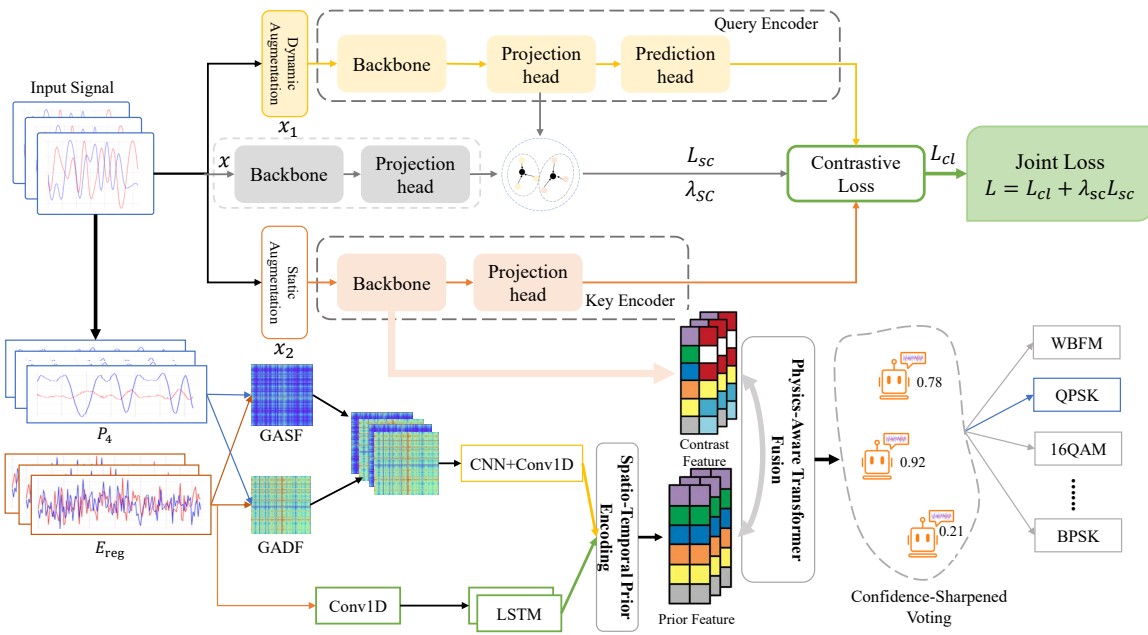

*Figure 1.* The overall architecture of DyCo-CL. It features a Signal-Adaptive Swin Backbone optimized via Dynamic-Consistency Pre-training (utilizing VAA and SC loss). The learned features are subsequently enhanced by a Hierarchical Hybrid Knowledge Fusion module that integrates spatio-temporal physical priors.

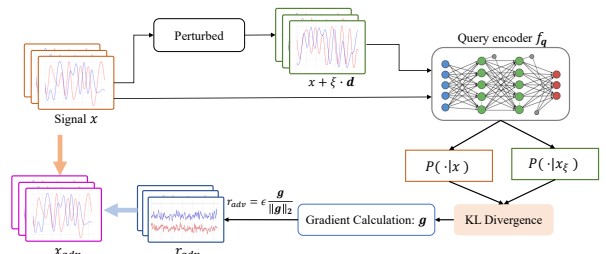

*Figure 2.* The generation process of Virtual Adversarial Augmentation.

We seek the optimal perturbation $\mathbf{r}^*$ within an $\epsilon$-ball that maximizes this divergence:

$$\mathbf{r}^* = \arg\max_{\|\mathbf{r}\|_2 \leq \epsilon} \mathcal{J}(\mathbf{r}). \tag{8}$$

As derived in Appendix A.2, the optimal perturbation $\mathbf{r}^*$ aligns with the dominant eigenvector of the Hessian matrix of $\mathcal{J}(\mathbf{r})$ at $\mathbf{r} = \mathbf{0}$. As illustrated in Figure 2, we approximate this direction efficiently via one-step Power Iteration.

Specifically, starting from a random unit vector $\mathbf{d}$, we estimate the Hessian-vector product using a finite-difference approximation with a small scalar $\xi$:

$$\mathbf{g} = \nabla_{\mathbf{r}} \mathcal{J}(\mathbf{r})\big|_{\mathbf{r}=\xi\mathbf{d}}, \quad \mathbf{x}_{adv} = \mathbf{x} + \epsilon \frac{\mathbf{g}}{\|\mathbf{g}\|_2}, \tag{9}$$

where $\xi$ controls the magnitude of the probing perturbation.

**Standard Physical Augmentation.** The second view $\mathbf{x}_{weak}$ is generated by stochastically composing transformations from a domain-specific set $\mathcal{T}_{phy}$. This ensures invariance to common physical distortions; crucially, we restrict the transformation magnitude to a conservative regime to prevent the augmented samples from crossing class boundaries, thereby avoiding the semantic drift discussed in Sec. 3.2.2. Detailed formulations are provided in Appendix B.

#### 4.2.2. SEMANTIC CONSISTENCY REGULARIZATION

To counteract the risk of *semantic drift* inherent in VAA and enforce geometric stability, we propose a Semantic Consistency Loss ($\mathcal{L}_{SC}$). Crucially, this objective implicitly minimizes the encoder's *local Lipschitz constant*, promoting *intra-class compactness* and output stability.

We explicitly constrain the adversarial representation $\mathbf{z}_{adv} = f_{\boldsymbol{q}}(\mathbf{x}_{adv})$ to the local neighborhood of the anchor $\mathbf{z} = f_{\boldsymbol{q}}(\mathbf{x})$:

$$\mathcal{L}_{SC} = 1 - \frac{\mathrm{sg}(\mathbf{z})^\top \mathbf{z}_{adv}}{\|\mathrm{sg}(\mathbf{z})\|_2 \|\mathbf{z}_{adv}\|_2}, \tag{10}$$

where $\mathrm{sg}(\cdot)$ denotes the stop-gradient operator.

**Geometric Interpretation.** The stop-gradient fixes $\mathbf{z}$ as a semantic centroid. $\mathcal{L}_{SC}$ exerts a restoring force against VAA-induced drift. As we rigorously prove in Sec. 5, this mechanism mathematically functions as a spectral regularizer, ensuring the geometric stability of the backbone.

**Algorithm 1** DyCo-AMR Pre-training Algorithm

**Input:** $f_q$, $f_k$ (encoders), $m$ (momentum), $\tau$ (temp), $\epsilon$, $\lambda_{sc}$

**for** each minibatch $\mathbf{x}$ in Dataset **do**

    $\mathbf{x}_{weak} = \text{PhysAug}(\mathbf{x})$,    $\mathbf{x}_{adv} = \text{VAA}(\mathbf{x}, f_q, \epsilon)$

    $q \leftarrow f_q(\mathbf{x}_{adv})$,    $k \leftarrow f_k(\mathbf{x}_{weak})$       *(k: no grad)*

    $l_{pos} = q \cdot k^{+}$,    $l_{neg} = q \cdot k^{-}$

    $\mathcal{L}_{NCE} = -\log \dfrac{\exp(l_{pos}/\tau)}{\exp(l_{pos}/\tau) + \sum \exp(l_{neg}/\tau)}$

    $z \leftarrow f_q^{\text{proj}}(\mathbf{x})$       *(projection output, no grad)*

    $z_{adv} \leftarrow f_q^{\text{proj}}(\mathbf{x}_{adv})$       *(projection output)*

    $\mathcal{L}_{SC} = 1 - \dfrac{z \cdot z_{adv}}{\|z\|\|z_{adv}\|}$

    $\mathcal{L}_{total} = \mathcal{L}_{NCE} + \lambda_{sc} \cdot \mathcal{L}_{SC}$

    Update $f_q$ via Backprop: $\nabla \mathcal{L}_{total}$

    Update $f_k$ via Momentum: $\theta_k \leftarrow m\theta_k + (1-m)\theta_q$

**end for**

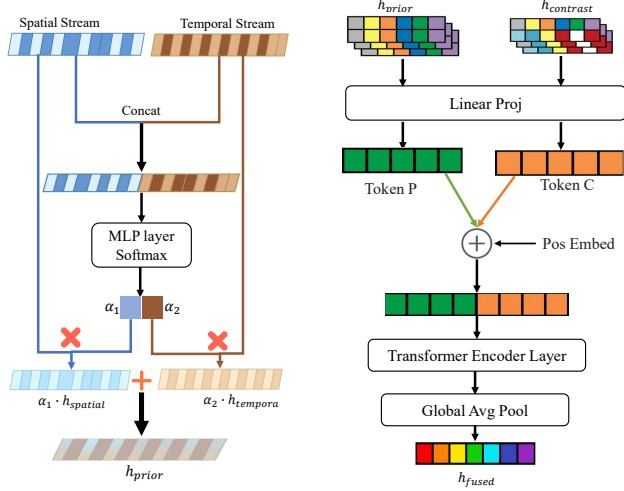

*(a)* Stage 1: Spatio-Temporal Prior Encoding

*(b)* Stage 2: Physics-Aware Transformer Fusion

*Figure 4.* Architecture of the Fusion Module.

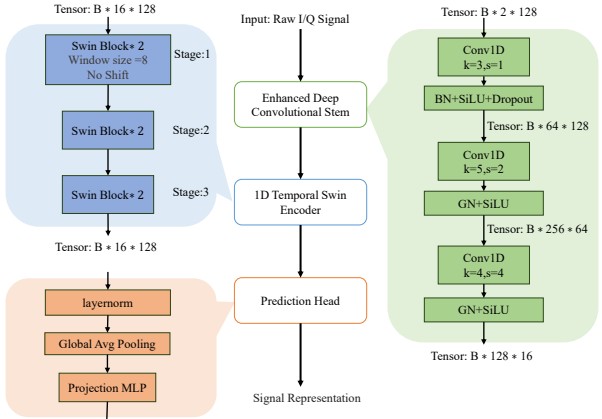

*Figure 3.* Structure of the Signal-Adaptive Swin Backbone.

### 4.2.3. OVERALL PRE-TRAINING OBJECTIVE

The final objective combines the instance-discrimination contrastive loss ($\mathcal{L}_{NCE}$) with semantic consistency. The total loss is defined as:

$$\mathcal{L}_{total} = \mathcal{L}_{NCE}(\mathbf{x}_{adv}, \mathbf{x}_{weak}) + \lambda_{sc} \cdot \mathcal{L}_{SC}(\mathbf{x}, \mathbf{x}_{adv}), \quad (11)$$

where $\lambda_{sc}$ balances representation diversity and semantic fidelity. The training procedure is summarized in Algorithm 1.

## 4.3. Signal-Adaptive Swin Backbone

Standard ViTs lack local inductive biases, while 2D Swin Transformers suffer from dimensionality mismatch and spectral instability. To address these, we propose a *Signal-Adaptive Swin Backbone* with a convolutional stem and 1D structural adaptation.

### 4.3.1. DEEP CONVOLUTIONAL STEM

To mitigate the noise sensitivity of linear embeddings, we design a hierarchical *Deep Convolutional Stem* (Fig. 3) comprising three stacked 1D convolution layers. This module acts as a learnable low-pass filter to suppress high-frequency artifacts. Furthermore, its overlapping receptive fields enable *soft tokenization*, preserving phase continuity across token boundaries while progressively downsampling temporal resolution.

### 4.3.2. 1D SWIN ENCODER

We partition the sequence $\mathbf{z} \in \mathbb{R}^{L \times D}$ into non-overlapping 1D windows of size $M$ and compute local self-attention:

$$\text{Attention}(Q, K, V) = \text{Softmax}\left(\frac{QK^{\top}}{\sqrt{d}} + \mathbf{B}_{1D}\right)V. \quad (12)$$

Leveraging the deep stem's receptive field for inter-window information exchange, we employ a *fixed window strategy* without shifting. Crucially, this non-overlapping partition enforces a block-diagonal Jacobian structure, structurally bounding the Lipschitz constant to ensure spectral stability (as justified in Sec. 5).

## 4.4. Hierarchical Hybrid Knowledge Fusion

To counteract VAA-induced *semantic drift* in few-shot regimes, we introduce a *Hierarchical Hybrid Knowledge Fusion* module (Fig. 4). This two-stage mechanism bridges the semantic gap by anchoring the manifold with physical priors.

### 4.4.1. STAGE 1: SPATIO-TEMPORAL PRIOR ENCODING

We first synthesize a robust physical descriptor $\mathbf{h}_{prior}$ to serve as a semantic reference. We extract two invariants: the *Fourth-Order Cycle Spectrum* ($\mathbf{P}_4$) for cyclostationary signatures, and the *PSD-Regularized Envelope* ($\mathbf{E}_{reg}$) for amplitude stability (see Appendix C). These features are processed via a dual-stream encoder: a *Spatial Stream* (GAF + 2D CNN) and a *Temporal Stream* (Bi-LSTM). A learnable gating network dynamically fuses them:

$$[\alpha_1, \alpha_2] = \text{Softmax}(\text{MLP}([\mathbf{h}_{spatial}; \mathbf{h}_{temporal}])). \quad (13)$$

$$\mathbf{h}_{prior} = \text{FC}(\alpha_1\mathbf{h}_{spatial} + \alpha_2\mathbf{h}_{temporal}). \quad (14)$$

### 4.4.2. STAGE 2: PHYSICS-AWARE TRANSFORMER FUSION

To enforce physical consistency, we employ a Transformer to model the non-linear interaction between the data-driven feature $\mathbf{h}_{contrast}$ and the physical prior $\mathbf{h}_{prior}$. We construct a composite sequence $\mathbf{T}$ with learnable type embeddings $\mathbf{E}_{type}$:

$$\mathbf{T} = [\text{Linear}(\mathbf{h}_{prior}); \text{Linear}(\mathbf{h}_{contrast})] + \mathbf{E}_{type}. \quad (15)$$

Through self-attention, the "Physics Token" acts as a stable query to dynamically calibrate the "Data Token", correcting potential semantic drift. The refined features are aggregated via GAP to yield $\mathbf{h}_{fused}$.

Finally, to mitigate variance in few-shot settings, we employ a *confidence-aware ensemble* of $K = 3$ heads with a sharpening mechanism:

$$\hat{\mathbf{y}} = \text{Normalize}\left(\sum_{k=1}^{K}[\text{Softmax}(\text{MLP}_k(\mathbf{h}_{fused}))]^2\right). \quad (16)$$

This quadratic weighting suppresses uncertain predictions, filtering out noise from ambiguous classifiers.

## 5. Theoretical Analysis: Geometric Stability via Structure and Optimization

In this section, we provide a theoretical justification for DyCo-CL. We demonstrate that our framework stabilizes the learning process through two complementary mechanisms: *structural constraints* (via Backbone) and *spectral optimization* (via Loss).

### 5.1. Structural Constraint via Signal-Adaptive Swin

Standard global attention yields an unbounded Jacobian $\|\mathbf{J}_{global}\|_2 \propto \sqrt{L}$. In contrast, our fixed-window strategy enforces a block-diagonal Jacobian $\mathbf{J}_{fixed}$:

$$\mathbf{J}_{fixed} = \text{diag}(\mathbf{J}_1, \ldots, \mathbf{J}_{N_w}). \quad (17)$$

The global Lipschitz constant is thus determined solely by the local window capacity:

$$\|\mathbf{J}_{fixed}\|_2 = \max_k \|\mathbf{J}_k\|_2 = \max_k \sigma_{max}(\mathbf{J}_k). \quad (18)$$

Since window size $M$ is fixed, this decouples the spectral norm from length $L$ (Proof in Appendix D). By bounding the attention mechanism (the primary instability source), this design ensures global stability despite local stem dependencies, establishing a prerequisite for the optimization below.

### 5.2. Optimization: DyCo-CL as Spectral Regularization

With the structural bound in place, DyCo-CL minimizes the effective Lipschitz constant via a Min-Max optimization strategy.

**Geometric Equivalence.** While our implementation uses Cosine distance ($\mathcal{L}_{SC}$), we base our theory on the squared Euclidean distance. Since representations are projected onto the unit hypersphere ($\|\mathbf{z}\|_2 = \|\mathbf{z}_{adv}\|_2 = 1$), these objectives are strictly equivalent. As proved in Appendix E.1, minimizing the cosine loss is mathematically identical to minimizing the Euclidean error:

$$\mathcal{L}_{SC} = 1 - \mathbf{z}^\top\mathbf{z}_{adv} = \frac{1}{2}\|\mathbf{z} - \mathbf{z}_{adv}\|_2^2. \quad (19)$$

Therefore, we analyze the following surrogate objective without loss of generality:

$$\min_\theta \mathcal{L}_{SC} \cong \min_\theta \mathbb{E}_{\mathbf{x}}\left[\max_{\|\mathbf{r}\|\leq\epsilon}\frac{1}{2}\|f_\theta(\mathbf{x}) - f_\theta(\mathbf{x} + \mathbf{r})\|_2^2\right]. \quad (20)$$

This surrogate is well-posed, as its inner maximization is asymptotically equivalent to the VAA objective of maximizing KL-divergence (see Appendix E.1 for a formal proof).

Applying a first-order Taylor expansion (valid for small perturbation radii $\epsilon \ll 1$; see derivation in Appendix E.2), the inner maximization becomes a Rayleigh quotient problem:

$$\max_{\|\mathbf{r}\|\leq\epsilon}\|\mathbf{J}_\theta(\mathbf{x})\mathbf{r}\|_2^2 = \epsilon^2\sigma_{max}^2(\mathbf{J}_\theta(\mathbf{x})). \quad (21)$$

Thus, the objective simplifies to minimizing the spectral norm:

$$\min_\theta \mathbb{E}_{\mathbf{x}}\left[\epsilon^2 \cdot \sigma_{max}^2(\mathbf{J}_\theta(\mathbf{x}))\right]. \quad (22)$$

**Theorem 5.1** (Implicit Spectral Regularization). *As the perturbation magnitude $\epsilon \to 0$, minimizing the Semantic Consistency loss under VAA is equivalent to minimizing the expected squared local Lipschitz constant, i.e.,*

$$\min_\theta \mathcal{L}_{SC} \iff \min_\theta \mathbb{E}_{\mathbf{x}}[K_f(\mathbf{x})^2]. \quad (23)$$

## 5.3. Generalization Bound for Few-Shot Learning

We link this regularization to generalization via statistical learning theory (Sokolic et al., 2017)(derivation provided in Appendix E.5). The generalization error $\mathcal{E}_{gen}$ is bounded by:

$$\mathcal{E}_{gen} \leq \hat{\mathcal{E}} + \mathcal{O}\left(\frac{\mathbb{E}_{\mathbf{x}\sim\mathcal{D}}\left[\sigma_{max}(\mathbf{J}_\theta(\mathbf{x}))\right]}{\sqrt{N}}\right). \quad (24)$$

In few-shot scenarios (small $N$), the error is dominated by the expected Lipschitz constant $\mathbb{E}_{\mathbf{x}\sim\mathcal{D}}[K_f(\mathbf{x})]$. By explicitly minimizing this quantity (via minimizing $\sigma_{max}$) in DyCo-CL, we tighten the bound, ensuring robust transferability.

# 6. Experiments

## 6.1. Experimental Setup

**Datasets.** We evaluate our framework on two standard benchmarks. RML2016.10a comprises 11 modulation schemes with Signal-to-Noise Ratios (SNRs) ranging from -20dB to 18dB in 2dB steps. To test scalability, we also employ the larger RML2018.01a, which includes 24 modulation types across an SNR range of -20dB to 30dB. Both datasets consist of raw complex-valued I/Q sequences.

**Implementation.** Pre-training runs for 50 epochs (AdamW, lr=$3e^{-4}$). We set VAA radius $\epsilon = 0.3$, power-iteration step size $\xi = 10^{-6}$, and consistency weight $\lambda_{sc} = 0.6$. For few-shot settings, we sample $N \in \{1, 2, 5, 10\}$ instances per class per SNR. Sensitivity analysis is provided in Appendix F.

## 6.2. Baselines

We compare DyCo-CL against six state-of-the-art semi-supervised methods: *CMSSAN* (Kong et al., 2025), *AMC-CNN* (Tekbıyık et al., 2020), *ResNet50-MoCo* (Davaslioglu et al., 2023), *EET-MoCo* (Chen et al., 2025), *SSCL-AMC* (Cai et al., 2025), and *APFS* (Bai et al., 2024). Unless otherwise specified, all SSL baselines are re-implemented by us and pre-trained on the same data using identical training protocols and compute budgets as DyCo-CL to ensure fair comparison.

## 6.3. Main Results

**Few-Shot Performance.** As shown in Fig. 5, DyCo-CL consistently outperforms all baselines. Notably, in the extreme $N = 1$ setting, it achieves 43.84% accuracy, surpassing the SOTA by 6.27%. This aligns with our theoretical analysis (Sec. 5): by minimizing the Lipschitz constant, DyCo-CL tightens the generalization bound, enabling robust transfer even with minimal supervision.

**Robustness Across SNRs.** Fig. 6 shows that DyCo-CL

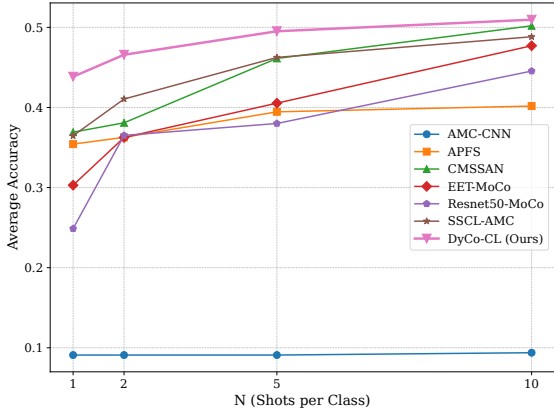

*Figure 5.* Classification accuracy comparison with varying $N$ on RML2016.10a.

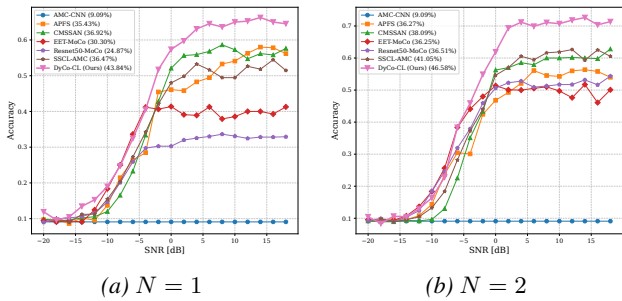

*(a)* $N = 1$        *(b)* $N = 2$

*Figure 6.* Accuracy vs. SNR comparison under low-data regimes on RML2016.10a.

consistently outperforms prior methods across SNRs. On RML2016.10a, it exceeds the SOTA by 7.7% at 10 dB ($N = 1$), and on RML2018.01a (Fig. 7) maintains a 7.54% margin for SNR > 6 dB. Notably, DyCo-CL exhibits an earlier performance inflection at low SNRs, indicating stronger noise robustness, which we attribute to the Signal-Adaptive Backbone and its Deep Convolutional Stem that suppresses high-frequency noise prior to tokenization.

## 6.4. Ablation Studies

We conduct subtractive ablations on RML2016.10a ($N = 1$) to isolate the effects of the backbone, VAA, and fusion design. The adaptive fusion is compared against simple concatenation baselines (Table 1).

**Analysis.** Table 1 confirms the synergy of our components: *(1) Manifold Expansion.* Removing VAA causes the largest drop ($-8.90\%$), proving that adversarial exploration is essential to overcome the *concentration of measure* and avoid trivial solution collapse. *(2) Structural Stability.* The Swin backbone outperforms ResNet18 ($-4.39\%$). Its block-diagonal Jacobian (Sec. 5.1) acts as a structural stabilizer, allowing the model to withstand high-energy VAA perturbations. *(3) Anchoring Semantics.* Simple concatenation

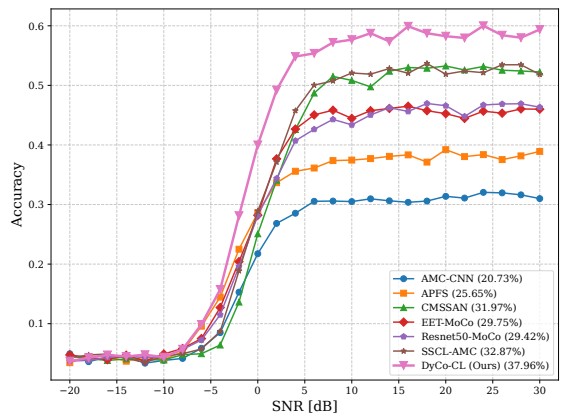

*Figure 7.* Accuracy vs. SNR comparison on RML2018.01a (10-shot).

*Table 1.* Ablation study on RML2016.10a ($N = 1$).

| Category | Model Variant | Acc (%) | $\Delta$ |
|---|---|---|---|
| **Full Method** | **DyCo-CL** | **43.84** | - |
| *Backbone & Module* | w/o Dynamic-Consistency | 34.94 | $-8.90$ |
| | w/o Swin (ResNet18) | 39.45 | $-4.39$ |
| *Fusion Strategy* | Stage I $\rightarrow$ Concat | 39.47 | $-4.37$ |
| | Stage II $\rightarrow$ Concat | 40.12 | $-3.72$ |
| | All Stages $\rightarrow$ Concat | 38.32 | $-5.52$ |

fails ($-5.52\%$). Our adaptive fusion counteracts *semantic drift* by explicitly anchoring the expanded manifold with physical priors, rather than just merging features.

### 6.5. Efficiency Analysis

DyCo-CL is highly efficient, achieving a practical sweet spot between robustness and edge-deployment constraints. Compared to the robust baseline SSCL-AMC, it reduces FLOPs by over $2.5\times$ while simultaneously tripling inference throughput. Furthermore, with only 1.44M parameters ($\approx 5.8$ MB), our model is over $16\times$ smaller than standard architectures like ResNet50-MoCo, and its 0.60 ms latency meets the sub-millisecond demands of real-time 5G applications. A comprehensive analysis is provided in Appendix G.

### 6.6. Qualitative Analysis

**Confusion Matrix Analysis.** Fig. 10 visualizes the classification behavior on the RML2016.10A dataset (10dB). DyCo-CL achieves distinct separation for phase-sensitive signals (e.g., 8PSK vs. QPSK), confirming the backbone's phase-preservation capability. Additional analysis on the RML2018.01A dataset is provided in the Appendix H.

**Feature Visualization (t-SNE).** Fig. 9 visualizes feature distributions on RML2016.10A and RML2018.01A. DyCo-CL consistently achieves high *intra-class compactness* and

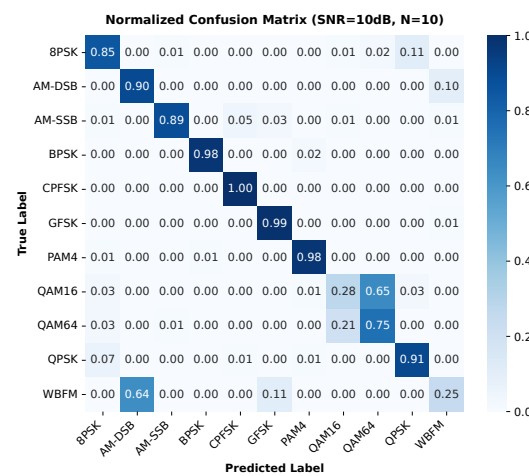

*Figure 8. Confusion Matrix ($N = 10$, 10dB) on RML2016.10A.*

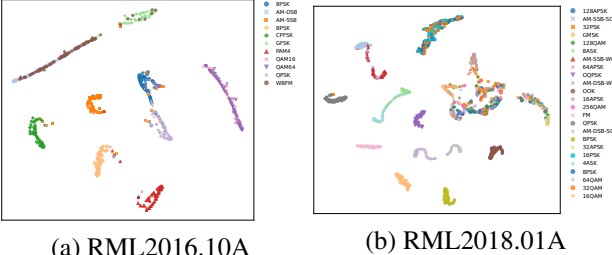

(a) RML2016.10A  (b) RML2018.01A

*Figure 9. t-SNE Visualization ($N = 10$, 10dB).* Feature distributions on RML2016.10A (Left) and RML2018.01A (Right).

clear *inter-class margins*, even with the increased complexity of the 2018 dataset. This validates that our spectral regularization effectively stabilizes the signal manifold across different data scales.

## 7. Conclusion

In this work, we address *concentration of measure*, *spectral instability*, and *semantic drift* via DyCo-CL. This framework couples VAA with Implicit Spectral Regularization to enforce optimization stability, while our Signal-Adaptive Swin structurally bounds the Lipschitz constant. Anchored by Hybrid Fusion, DyCo-CL establishes a new few-shot SOTA. Future work will explore open-set recognition.

## Impact Statement

This paper presents work whose goal is to advance the field of Machine Learning. There are many potential societal consequences of our work, none which we feel must be specifically highlighted here.

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

# A. Theoretical Analysis and Proofs

In this section, we provide rigorous mathematical derivations to support the geometric motivations presented in Section 3.2 and the optimality of the VAA strategy in Section 4.2.1.

## A.1. Proof of the Orthogonality in High Dimensions

**Proposition 1 (Asymptotic Orthogonality).** *Let $\mathbf{v} \in \mathbb{R}^D$ be a fixed unit vector and $\mathbf{r} \sim \mathcal{N}(0, \sigma^2 \mathbf{I}_D)$ be an isotropic random perturbation. As $D \to \infty$, $\mathbf{r}$ becomes orthogonal to $\mathbf{v}$ almost surely.*

*Proof.* Due to the rotational invariance of the isotropic Gaussian distribution, the projection of $\mathbf{r}$ onto any fixed unit vector $\mathbf{v}$ follows a univariate Gaussian distribution. Let $z = \mathbf{r}^\top \mathbf{v}$. Then:

$$z \sim \mathcal{N}(0, \sigma^2). \tag{25}$$

The squared norm $\|\mathbf{r}\|^2$ follows a scaled Chi-square distribution, i.e., $\|\mathbf{r}\|^2 / \sigma^2 \sim \chi_D^2$. According to the concentration properties of Chi-square variables, for large $D$, the norm concentrates sharply around its mean:

$$\|\mathbf{r}\| \approx \sigma \sqrt{D}. \tag{26}$$

More formally, for any $\eta > 0$, the probability of deviation decays exponentially: $P(|\|\mathbf{r}\| - \sigma\sqrt{D}| \geq \eta \sigma \sqrt{D}) \leq 2e^{-cD\eta^2}$. Thus, we can approximate the denominator $\|\mathbf{r}\|$ by the deterministic value $\sigma\sqrt{D}$ with high probability.

The cosine similarity $\cos\theta$ between the perturbation $\mathbf{r}$ and the direction $\mathbf{v}$ is given by:

$$\cos\theta = \frac{\mathbf{r}^\top \mathbf{v}}{\|\mathbf{r}\|} = \frac{z}{\|\mathbf{r}\|}. \tag{27}$$

Substituting the concentration approximation $\|\mathbf{r}\| \approx \sigma\sqrt{D}$:

$$\cos\theta \approx \frac{z}{\sigma\sqrt{D}}. \tag{28}$$

Since $z \sim \mathcal{N}(0, \sigma^2)$, the normalized variable $z/\sigma$ follows a standard normal distribution $\mathcal{N}(0, 1)$. Let $\delta > 0$ be a small angular tolerance. We apply the standard Gaussian tail bound $P(|X| \geq t) \leq 2\exp(-t^2/2)$ with $t = \delta\sqrt{D}$:

$$P(|\cos\theta| \geq \delta) \approx P\left(\left|\frac{z}{\sigma}\right| \geq \delta\sqrt{D}\right) \leq 2\exp\left(-\frac{D\delta^2}{2}\right). \tag{29}$$

As $D \to \infty$, the exponent $-\frac{D\delta^2}{2} \to -\infty$, and thus $P(|\cos\theta| \geq \delta) \to 0$. This proves that the perturbation $\mathbf{r}$ is orthogonal to the sensitive direction $\mathbf{v}$ almost surely. $\square$

**Remark (Finite Dimension Analysis).** While Proposition 1 establishes asymptotic orthogonality, we verify its validity for the specific dimension of AMR signals ($D = 256$). Typically, a perturbation is considered effective if it has a non-negligible projection on the gradient, e.g., $|\cos\theta| \geq 0.2$ (corresponding to an angle $\leq 78.5°$). Substituting $D = 256$ and $\delta = 0.2$ into Eq. (4):

$$P(|\cos\theta| \geq 0.2) \leq 2\exp\left(-\frac{256 \times 0.2^2}{2}\right) = 2e^{-5.12} \approx 0.012. \tag{30}$$

This indicates that even in finite dimensions, over $98.8\%$ of isotropic random perturbations are effectively orthogonal to the sensitive direction. Thus, the concentration of measure phenomenon remains the dominant geometric constraint in our setting.

### A.2. Optimality of Virtual Adversarial Augmentation

Here we derive why the VAA update rule targets the most sensitive direction of the model.

**Problem Setup.** We seek a perturbation $\mathbf{r}$ with $\|\mathbf{r}\| \leq \epsilon$ that maximizes the KL divergence between the output distributions of the clean input $\mathbf{x}$ and the perturbed input $\mathbf{x} + \mathbf{r}$:

$$\mathbf{r}^* = \arg\max_{\|\mathbf{r}\| \leq \epsilon} \mathcal{J}(\mathbf{r}), \quad \text{where } \mathcal{J}(\mathbf{r}) = \mathrm{KL}[P(\cdot|\mathbf{x}) \,\|\, P(\cdot|\mathbf{x} + \mathbf{r})]. \tag{31}$$

**Taylor Expansion.** Since $\mathcal{J}(\mathbf{0}) = 0$ (divergence with itself is zero) and $\mathcal{J}(\mathbf{r})$ is minimized at $\mathbf{r} = \mathbf{0}$, the first-order gradient $\nabla_\mathbf{r}\mathcal{J}(\mathbf{0})$ is also $\mathbf{0}$. We perform a second-order Taylor expansion around $\mathbf{r} = \mathbf{0}$:

$$\mathcal{J}(\mathbf{r}) \approx \mathcal{J}(\mathbf{0}) + \nabla_\mathbf{r}\mathcal{J}(\mathbf{0})^\top \mathbf{r} + \frac{1}{2}\mathbf{r}^\top \mathbf{H}(\mathbf{x})\mathbf{r} = \frac{1}{2}\mathbf{r}^\top \mathbf{H}(\mathbf{x})\mathbf{r}, \tag{32}$$

where $\mathbf{H}(\mathbf{x}) = \nabla_\mathbf{r}^2 \mathcal{J}(\mathbf{r})|_{\mathbf{r}=0}$ is the Hessian matrix of the KL divergence with respect to the input.

**Eigenvector Alignment.** The optimization problem simplifies to maximizing the quadratic form:

$$\mathbf{r}^* \approx \arg\max_{\|\mathbf{r}\| \leq \epsilon} \frac{1}{2}\mathbf{r}^\top \mathbf{H}(\mathbf{x})\mathbf{r}. \tag{33}$$

From linear algebra, the vector $\mathbf{r}$ that maximizes $\mathbf{r}^\top \mathbf{H}\mathbf{r}$ subject to a norm constraint is the **dominant eigenvector** (the eigenvector corresponding to the largest eigenvalue) of the Hessian $\mathbf{H}$. Let $\mathbf{u}_1$ be this unit eigenvector. Then:

$$\mathbf{r}^* = \epsilon \mathbf{u}_1. \tag{34}$$

**Power Iteration Approximation.** Computing the full Hessian $\mathbf{H}$ is computationally expensive ($\mathcal{O}(D^2)$). The Power Iteration method finds $\mathbf{u}_1$ by iteratively computing $\mathbf{d}_{t+1} \leftarrow \mathbf{H}\mathbf{d}_t$. In our VAA implementation, we approximate the Hessian-vector product $\mathbf{H}\mathbf{d}$ using the finite difference of gradients:

In practice, the Hessian-vector product $\mathbf{H}\mathbf{d}$ is approximated using a finite-difference scheme with a small scalar $\xi > 0$:

$$\mathbf{H}\mathbf{d} \approx \frac{\nabla_\mathbf{r}\mathcal{J}(\mathbf{r})|_{\mathbf{r}=\xi\mathbf{d}} - \nabla_\mathbf{r}\mathcal{J}(\mathbf{r})|_{\mathbf{r}=0}}{\xi} = \frac{\nabla_\mathbf{r}\mathcal{J}(\mathbf{r})|_{\mathbf{r}=\xi\mathbf{d}}}{\xi}, \tag{35}$$

where the second equality follows from $\nabla_\mathbf{r}\mathcal{J}(\mathbf{0}) = \mathbf{0}$.

By performing one step of this approximation, we effectively align the perturbation $\mathbf{r}$ with the dominant eigenvector of the local curvature, thereby targeting the direction where the model is most sensitive.

## B. Details of Standard Physical Augmentations

In the standard augmentation branch (generating $\mathbf{x}_{weak}$), we apply a set of domain-specific transformations $\mathcal{T}_{std}$ to the complex baseband signal $s[n] = I[n] + jQ[n]$. Each transformation is applied sequentially with a probability of $p = 0.5$. The specific formulations are as follows:

- **Random Rotation:** To simulate phase ambiguity caused by lack of synchronization, we rotate the I/Q constellation by a random angle $\theta$:
$$s'[n] = s[n] \cdot e^{j\theta}, \quad \theta \sim \mathcal{U}(0, \pi). \tag{36}$$

- **I/Q Flip:** To model spectrum inversion or hardware polarity mismatches, we randomly invert the sign of the in-phase or quadrature components:
$$s'[n] = \begin{cases} -\mathrm{Re}(s[n]) + j\mathrm{Im}(s[n]) & \text{flip I} \\ \mathrm{Re}(s[n]) - j\mathrm{Im}(s[n]) & \text{flip Q} \\ -s[n] & \text{flip Both} \end{cases} \tag{37}$$

- **Time Shift:** To simulate temporal synchronization errors, we cyclically shift the signal sequence by an integer delay $\delta$:

$$s'[n] = s[(n - \delta) \pmod{L}], \quad \delta \sim \mathcal{U}\{1, L/16\}. \tag{38}$$

- **AWGN Injection:** To enhance robustness against varying Signal-to-Noise Ratios (SNR), we inject complex additive white Gaussian noise:

$$s'[n] = s[n] + w[n], \quad w[n] \sim \mathcal{CN}(0, \sigma^2), \tag{39}$$

where the noise standard deviation $\sigma$ is uniformly sampled from $\mathcal{U}(0.01, 0.04)$.

- **Frequency Offset:** To simulate Carrier Frequency Offset (CFO) due to oscillator mismatch, we apply a linear phase progression:

$$s'[n] = s[n] \cdot e^{j2\pi\Delta f \frac{n}{L}}, \quad \Delta f \sim \mathcal{U}(-1, 1). \tag{40}$$

- **Amplitude Scaling:** To simulate channel fading or gain control variations, we scale the signal magnitude by a random factor $\alpha$:

$$s'[n] = \alpha \cdot s[n], \quad \alpha \sim \mathcal{U}(0.8, 1.2). \tag{41}$$

Finally, the augmented complex sequence $s'[n]$ is converted back to the real-valued matrix form $\mathbf{x}_{weak} \in \mathbb{R}^{2 \times L}$ for model input.

## C. Details of Physical Prior Extraction

In this section, we provide the detailed mathematical formulations for the expert features used in the Hierarchical Hybrid Knowledge Fusion module.

### C.1. Fourth-Order Cycle Spectrum ($\mathbf{P}_4$)

To capture cyclostationary signatures hidden in noise, we compute the magnitude spectrum of the fourth-power signal. Let $x[n]$ be the complex baseband signal of length $L$. We first compute the fourth power $x^4[n]$ and then apply the Discrete Fourier Transform (DFT):

$$X_4[k] = \sum_{n=0}^{L-1} (x[n])^4 e^{-j\frac{2\pi}{L}kn}. \tag{42}$$

The normalized feature vector $\mathbf{P}_4 \in \mathbb{R}^L$ is obtained by:

$$\mathbf{P}_4[k] = \frac{|X_4[k]|}{\max_k |X_4[k]| + \epsilon}. \tag{43}$$

This feature amplifies phase symmetries inherent in high-order constellations (e.g., QAM, PSK).

### C.2. PSD-Regularized Envelope ($\mathbf{E}_{reg}$)

To characterize amplitude stability robust to SNR variations, we derive a normalized envelope feature. First, we compute the instantaneous amplitude $r[n] = \sqrt{I[n]^2 + Q[n]^2}$. We then compute the zero-centered normalized envelope $r_{cn}[n]$:

$$r_{cn}[n] = \frac{r[n] - \bar{r}}{\max |r[n] - \bar{r}| + \epsilon}, \tag{44}$$

where $\bar{r}$ is the mean amplitude. Next, we compute the peak Power Spectral Density ($\gamma_{max}$) of $r_{cn}$:

$$\gamma_{max} = \frac{1}{L} \max_k \left| \sum_{n=0}^{L-1} r_{cn}[n] e^{-j\frac{2\pi}{L}kn} \right|^2. \tag{45}$$

Finally, the regularized envelope feature $\mathbf{E}_{reg} \in \mathbb{R}^L$ is derived by normalizing the original amplitude by $\gamma_{max}$:

$$\mathbf{E}_{reg}[n] = \text{Norm}\left( \frac{r[n]}{\gamma_{max} + \epsilon} \right). \tag{46}$$

## C.3. Gramian Angular Fields (GAF)

In the Spatial Stream, we transform the 1D features into 2D manifolds. Given a normalized time series $\mathbf{x} = \{x_1, \ldots, x_L\}$, we first rescale it to $[-1, 1]$ and convert it to polar coordinates via $\phi_i = \arccos(x_i)$. The Gramian Angular Sum Field (GASF) and Difference Field (GADF) are defined as:

$$\text{GASF}_{ij} = \cos(\phi_i + \phi_j), \quad \text{GADF}_{ij} = \sin(\phi_i - \phi_j). \tag{47}$$

These 2D maps preserve temporal correlations in a spatial structure suitable for CNN processing.

# D. Spectral Stability of Signal-Adaptive Swin

We provide the formal proof for Proposition 1, demonstrating that the proposed fixed-window attention mechanism structurally bounds the Lipschitz constant.

**Definition (Local Lipschitz Constant).** For a function $f : \mathcal{X} \to \mathcal{Y}$, the local Lipschitz constant at input $\mathbf{x}$ is defined as the spectral norm of its Jacobian matrix $\mathbf{J}_f(\mathbf{x}) = \partial f(\mathbf{x})/\partial \mathbf{x}$. We denote this as:

$$K_f(\mathbf{x}) \triangleq \|\mathbf{J}_f(\mathbf{x})\|_2 = \sigma_{max}(\mathbf{J}_f(\mathbf{x})), \tag{48}$$

where $\sigma_{max}(\cdot)$ denotes the largest singular value.

**Proposition 1 (Structural Boundedness).** *Let $f_{global}$ be a standard global self-attention layer, and $f_{fixed}$ be the proposed fixed-window attention layer with window size $M$. The Lipschitz constant of $f_{fixed}$ is strictly bounded by a constant $C_M$ dependent only on $M$, whereas $f_{global}$ is unbounded with respect to sequence length $L$.*

*Proof.* Let $\mathbf{X} \in \mathbb{R}^{L \times D}$ be the input sequence (which can be vectorized as $\mathbf{x} \in \mathbb{R}^{LD}$).

**1. Instability of Global Attention.** For standard dot-product attention, Kim et al. (Kim et al., 2021) proved that the Jacobian $\mathbf{J}_{global}$ is a dense matrix, and its spectral norm scales with the sequence length:

$$\sup_{\mathbf{X}} \sigma_{max}(\mathbf{J}_{global}(\mathbf{X})) = \mathcal{O}(\sqrt{L}). \tag{49}$$

As $L \to \infty$, the Lipschitz constant diverges, causing spectral instability.

**2. Stability of Fixed-Window Attention.** In our backbone, $\mathbf{X}$ is partitioned into $N_w = L/M$ non-overlapping windows $\{\mathbf{W}_k\}$. The function $f_{fixed}$ operates independently on each window. Consequently, the Jacobian $\mathbf{J}_{fixed}$ is strictly block-diagonal:

$$\mathbf{J}_{fixed} = \text{diag}(\mathbf{J}_1, \ldots, \mathbf{J}_{N_w}), \tag{50}$$

where $\mathbf{J}_k$ is the local Jacobian for the $k$-th window.

**3. Derivation of the Bound.** The largest singular value of a block-diagonal matrix is the maximum of the singular values of its blocks:

$$\sigma_{max}(\mathbf{J}_{fixed}) = \max_k \sigma_{max}(\mathbf{J}_k). \tag{51}$$

Let $C_M = \sup_{\mathbf{W}} \sigma_{max}(\mathbf{J}_\phi(\mathbf{W}))$ be the Lipschitz constant of the local window attention. Since $M$ is a fixed constant (e.g., $M = 16$) and $M \ll L$, $C_M$ is independent of $L$. Thus:

$$K_{f_{fixed}}(\mathbf{X}) = \sigma_{max}(\mathbf{J}_{fixed}) \leq C_M < \infty. \tag{52}$$

This proves that the Lipschitz constant is structurally bounded, ensuring geometric stability. $\square$

# E. Detailed Proof of Implicit Spectral Regularization

In this section, we prove that the Dynamic-Consistency objective functions as a spectral regularizer. We utilize the notation $K_f(\mathbf{x}) = \sigma_{max}(\mathbf{J}_f(\mathbf{x}))$ defined in Appendix D.

## E.1. Problem Setup and Theoretical Equivalences

The Semantic Consistency (SC) loss implemented in our algorithm minimizes the cosine distance between the anchor $\mathbf{z} = f_\theta(\mathbf{x})$ and the adversarial view $\mathbf{z}_{adv} = f_\theta(\mathbf{x} + \mathbf{r})$.

First, we establish the strict equivalence between this implemented loss and the Euclidean distance used for theoretical analysis. Since the contrastive representations are projected onto the unit hypersphere, we have $\|\mathbf{z}\|_2 = \|\mathbf{z}_{adv}\|_2 = 1$. Expanding the squared Euclidean distance:

$$\frac{1}{2}\|\mathbf{z} - \mathbf{z}_{adv}\|_2^2 = \frac{1}{2}(\mathbf{z} - \mathbf{z}_{adv})^\top(\mathbf{z} - \mathbf{z}_{adv}) \tag{53}$$

$$= \frac{1}{2}(\underbrace{\|\mathbf{z}\|_2^2}_{1} - 2\mathbf{z}^\top\mathbf{z}_{adv} + \underbrace{\|\mathbf{z}_{adv}\|_2^2}_{1}) \tag{54}$$

$$= 1 - \mathbf{z}^\top\mathbf{z}_{adv} \tag{55}$$

$$= \mathcal{L}_{SC}. \tag{56}$$

This derivation proves that minimizing the implemented Cosine loss is mathematically identical to minimizing the Euclidean displacement. Consequently, without loss of generality, we can formulate the optimization problem using the Euclidean norm to leverage its spectral properties for analysis.

Based on the loss equivalence, we analyze the following surrogate objective:

$$\min_\theta \mathbb{E}_\mathbf{x}\left[\max_{\|\mathbf{r}\|\leq\epsilon} \frac{1}{2}\|f_\theta(\mathbf{x}) - f_\theta(\mathbf{x} + \mathbf{r})\|_2^2\right]. \tag{57}$$

Next, we formally justify that the inner maximization of Eq. (57), which maximizes $L_2$ displacement, is asymptotically equivalent to the VAA objective used in our algorithm.

The VAA objective is to find a perturbation $\mathbf{r}_{adv}$ that maximizes the KL-divergence of the output distributions:

$$\mathbf{r}_{adv} = \arg\max_{\|\mathbf{r}\|_2\leq\epsilon} D_{KL}(P_\theta(\mathbf{x}) \,\|\, P_\theta(\mathbf{x} + \mathbf{r})). \tag{58}$$

For a small perturbation $\mathbf{r}$, the KL-divergence can be approximated by its second-order Taylor expansion:

$$D_{KL}(P_\theta(\mathbf{x}) \,\|\, P_\theta(\mathbf{x} + \mathbf{r})) \approx \frac{1}{2}\mathbf{r}^\top\mathbf{F}(\mathbf{x})\mathbf{r}, \tag{59}$$

where $\mathbf{F}(\mathbf{x})$ is the Fisher Information Matrix (FIM) with respect to the input $\mathbf{x}$.

A key property connecting the FIM to the representation geometry is its relationship with the Jacobian of the feature map, $\mathbf{J}_\theta(\mathbf{x}) = \nabla_\mathbf{x} f_\theta(\mathbf{x})$. For distributions where the representation $f_\theta(\mathbf{x})$ acts as the natural parameter (a common setup in contrastive learning), the FIM is directly proportional to the Gram matrix of the Jacobian's pushforward map (Pascanu et al., 2013). This leads to the approximation:

$$\mathbf{r}^\top\mathbf{F}(\mathbf{x})\mathbf{r} \propto \mathbf{r}^\top(\mathbf{J}_\theta(\mathbf{x})^\top\mathbf{J}_\theta(\mathbf{x}))\mathbf{r} = \|\mathbf{J}_\theta(\mathbf{x})\mathbf{r}\|_2^2. \tag{60}$$

Meanwhile, the objective analyzed in our main theoretical track is the maximization of the squared $L_2$ displacement. Using a first-order Taylor expansion, this is:

$$\max_{\|\mathbf{r}\|_2\leq\epsilon} \|f_\theta(\mathbf{x} + \mathbf{r}) - f_\theta(\mathbf{x})\|_2^2 \approx \max_{\|\mathbf{r}\|_2\leq\epsilon} \|\mathbf{J}_\theta(\mathbf{x})\mathbf{r}\|_2^2. \tag{61}$$

Comparing the two maximization problems, we see that both are approximately equivalent to finding the perturbation $\mathbf{r}$ that maximizes the Rayleigh quotient $\frac{\mathbf{r}^\top\mathbf{J}_\theta(\mathbf{x})^\top\mathbf{J}_\theta(\mathbf{x})\mathbf{r}}{\mathbf{r}^\top\mathbf{r}}$. The solution to this is the dominant eigenvector of $\mathbf{J}_\theta(\mathbf{x})^\top\mathbf{J}_\theta(\mathbf{x})$, which corresponds to the direction of the largest singular value of the Jacobian $\mathbf{J}_\theta(\mathbf{x})$.

Therefore, the adversarial direction found by VAA is asymptotically the same as the one that maximally displaces the feature representation in Euclidean space. This formally validates the analysis of the surrogate objective in Eq. (57).

### E.2. Step 1: Local Linearization

Using a first-order Taylor expansion around $\mathbf{x}$[1]:

$$f_\theta(\mathbf{x} + \mathbf{r}) \approx f_\theta(\mathbf{x}) + \mathbf{J}_\theta(\mathbf{x})\mathbf{r}. \tag{62}$$

The objective function approximates to:

$$\|f_\theta(\mathbf{x}) - f_\theta(\mathbf{x} + \mathbf{r})\|_2^2 \approx \|\mathbf{J}_\theta(\mathbf{x})\mathbf{r}\|_2^2 = \mathbf{r}^\top \mathbf{J}_\theta(\mathbf{x})^\top \mathbf{J}_\theta(\mathbf{x})\mathbf{r}. \tag{63}$$

### E.3. Step 2: Solving the Inner Maximization

The inner loop seeks the perturbation $\mathbf{r}^*$ that maximizes this quadratic form under $\|\mathbf{r}\|_2 \leq \epsilon$. This is a Rayleigh quotient problem. The maximum value is determined by the largest eigenvalue of $\mathbf{J}_\theta(\mathbf{x})^\top \mathbf{J}_\theta(\mathbf{x})$:

$$\max_{\|\mathbf{r}\| \leq \epsilon} \|\mathbf{J}_\theta(\mathbf{x})\mathbf{r}\|_2^2 = \epsilon^2 \lambda_{max}(\mathbf{J}_\theta(\mathbf{x})^\top \mathbf{J}_\theta(\mathbf{x})). \tag{64}$$

By definition, $\sqrt{\lambda_{max}(\mathbf{A}^\top \mathbf{A})} = \sigma_{max}(\mathbf{A})$. Thus:

$$\max_{\|\mathbf{r}\| \leq \epsilon} \|\mathbf{J}_\theta(\mathbf{x})\mathbf{r}\|_2^2 = \epsilon^2 \sigma_{max}^2(\mathbf{J}_\theta(\mathbf{x})). \tag{65}$$

### E.4. Step 3: Equivalence to Lipschitz Regularization

Substituting back into the outer minimization:

$$\min_\theta \mathbb{E}_\mathbf{x} \left[ \epsilon^2 \sigma_{max}^2(\mathbf{J}_\theta(\mathbf{x})) \right] \propto \min_\theta \mathbb{E}_\mathbf{x}[K_f(\mathbf{x})^2]. \tag{66}$$

This confirms that DyCo-CL explicitly minimizes the local Lipschitz constant.

### E.5. Step 4: Link to Generalization Bound

Following Sokolic et al. (Sokolic et al., 2017), the generalization error $\mathcal{E}_{gen}$ is bounded by the spectral norm of the Jacobian:

$$\mathcal{E}_{gen} \leq \hat{\mathcal{E}} + \mathcal{O}\left( \frac{\mathbb{E}_{\mathbf{x} \sim \mathcal{D}}\left[\sigma_{max}(\mathbf{J}_\theta(\mathbf{x}))\right]}{\sqrt{N}} \right). \tag{67}$$

In few-shot regimes (small $N$), minimizing $\sigma_{max}(\mathbf{J}_\theta(\mathbf{x}))$ (via DyCo-CL) is critical for reducing the generalization gap.

## F. Hyperparameter Sensitivity Analysis

In this section, we analyze the sensitivity of DyCo-CL to four critical hyperparameters: the semantic consistency weight ($\lambda_{sc}$), the VAA perturbation radius ($\epsilon$), the Swin Transformer window size ($M$), and the power iteration steps ($I_{iter}$). All experiments are conducted on RML2016.10a under the 1-shot setting.

**Remark on Window Size.** We observe an optimal trade-off at $M = 8$.

- **Too Large** ($M = 16$): Setting $M = 16$ (equal to feature resolution) degenerates into global attention. This degrades performance by 1.21% compared to $M = 8$, validating our claim that structural locality is needed for geometric stability.

- **Too Small** ($M < 8$): Conversely, overly narrow windows severely restrict the receptive field, preventing the capture of continuous signal patterns (e.g., modulation cycles).

---

[1]We assume the encoder $f_\theta$ is locally linear within the $\epsilon$-ball. While deep networks are globally non-linear, this first-order approximation is standard in adversarial training literature (Walters, 2015) to provide tractable geometric insights.

*Table 2.* **Sensitivity to Loss Hyperparameters.** Impact of consistency weight ($\lambda_{sc}$) and perturbation radius ($\epsilon$).

**(a) Impact of Consistency Weight ($\lambda_{sc}$)**

| Value | 0.0 | 0.1 | 0.2 | 0.3 | 0.4 | 0.5 | **0.6 (Default)** | 0.7 | 0.8 | 0.9 |
|---|---|---|---|---|---|---|---|---|---|---|
| Acc (%) | 38.38 | 38.43 | 39.30 | 41.58 | 41.96 | 41.78 | **43.84** | 41.04 | 42.44 | 40.91 |
| $\Delta$ | -5.46 | -5.41 | -4.54 | -2.26 | -1.88 | -2.06 | - | -2.8 | -1.4 | -2.93 |

**(b) Impact of VAA Perturbation Radius ($\epsilon$)**

| Value | 0.1 | 0.2 | **0.3 (Default)** | 0.4 | 0.5 |
|---|---|---|---|---|---|
| Acc (%) | 43.70 | 43.05 | **43.84** | 41.34 | 40.95 |
| $\Delta$ | -0.14 | -0.79 | - | -2.50 | -2.89 |

*Table 3.* **Sensitivity to Architecture and Compute.** Impact of window size ($M$) and power iteration steps ($I_{iter}$).

**(c) Impact of Swin Window Size ($M$)**

| Value | 1 | 2 | 4 | **8 (Default)** | 16 |
|---|---|---|---|---|---|
| Acc (%) | 39.29 | 39.15 | 38.10 | **43.84** | 42.63 |
| $\Delta$ | -4.55 | -4.69 | -5.74 | - | -1.21 |

**(d) Impact of Power Iteration Steps ($I_{iter}$)**

| Iterations | **1 (Default)** | 2 | 5 |
|---|---|---|---|
| Acc (%) | **43.84** | 41.71 | 40.82 |
| Time | **1.0$\times$** | 1.19$\times$ | 1.38$\times$ |
| $\Delta$ | - | -2.13 | -3.02 |

Thus, $M = 8$ balances structural stability with sufficient semantic context.

**Analysis of Power Iteration.** Table 3(d) shows that increasing $I_{iter}$ beyond 1 degrades performance ($-2.13\%$ at $I_{iter} = 2$). We attribute this to an *Over-Adversarial Effect*: multi-step iterations generate overly aggressive perturbations that cross decision boundaries, exacerbating *semantic drift*. In contrast, the single-step approximation ($I_{iter} = 1$) provides a coarse yet effective direction, enhancing robustness while preserving semantic fidelity. Thus, $I_{iter} = 1$ is optimal for both efficiency and stability.

## G. Detailed Complexity and Deployment Analysis

To comprehensively evaluate the feasibility of DyCo-CL for edge deployment, we extend our analysis beyond theoretical complexity (FLOPs/Params) to practical hardware indicators, including inference latency, storage footprint, and throughput. All efficiency experiments were conducted on a workstation equipped with an *AMD EPYC 9554 64-Core Processor* and a single *NVIDIA RTX 4090*, using PyTorch with FP32 precision, as detailed in Table 4.

**Storage Efficiency.** DyCo-CL is extremely lightweight. With only 1.44M parameters (occupying $\approx$5.8 MB), it is 16$\times$ smaller than the standard ResNet50-MoCo (94 MB), making it ideal for memory-constrained edge devices.

**Computational Trade-offs.** We observe a distinction between ultra-lightweight CNNs and robust architectures:

**Vs. Ultra-Lightweight (CMSSAN):** While CMSSAN offers extreme speed via shallow depth, it lacks the capacity for complex signal modeling. DyCo-CL prioritizes representational stability over raw speed.

**Vs. Robust Baselines (SSCL-AMC):** This is where DyCo-CL excels. Compared to its direct competitor SSCL-AMC, DyCo-CL reduces FLOPs by 2.5$\times$ (36.9 $\rightarrow$ 14.5 M) and triples the inference throughput (1672 vs. 498 samples/s).

**Real-Time Feasibility.** DyCo-CL achieves a latency of 0.60 ms, falling comfortably within the sub-millisecond scheduling

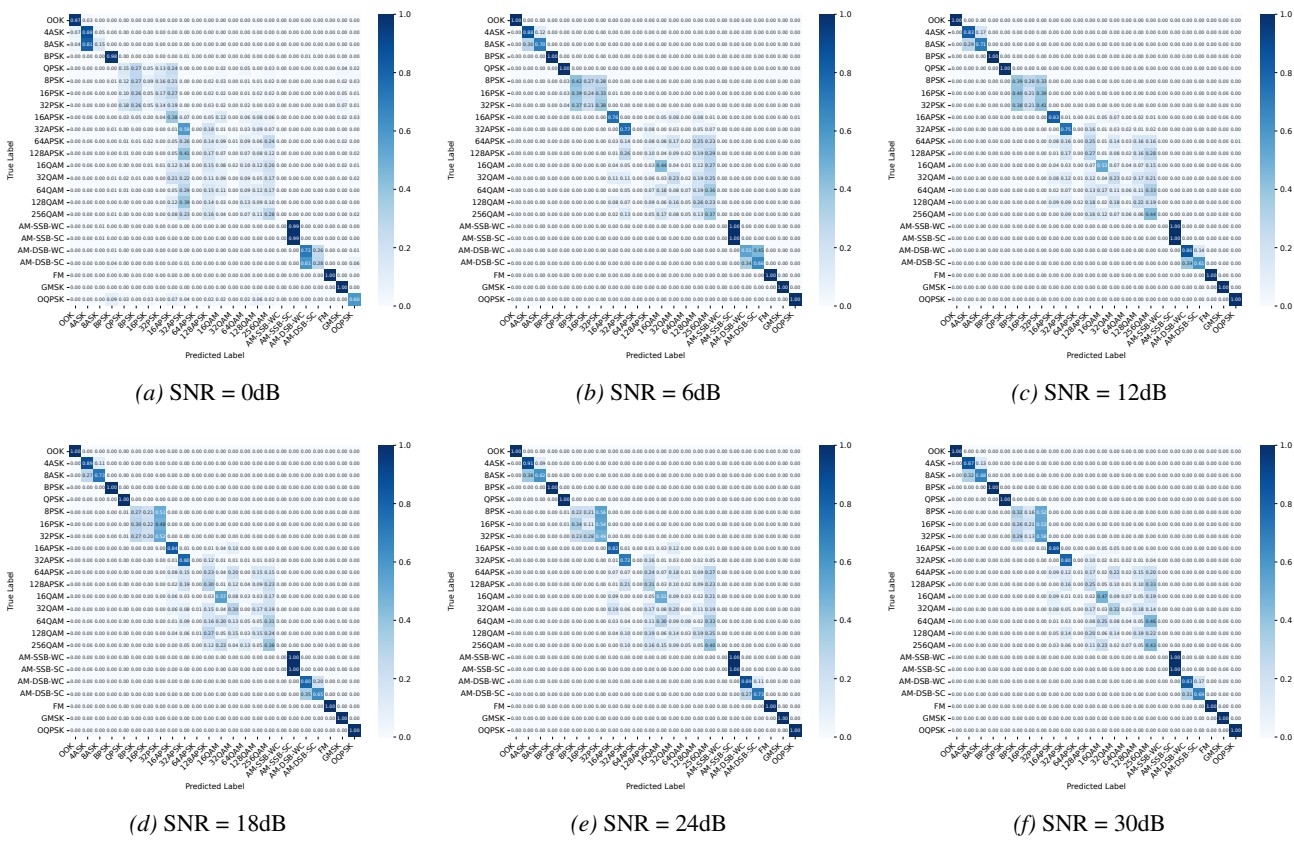

*Figure 10.* Confusion Matrices across different SNRs ($N = 10$). The 2×3 grid demonstrates the evolution of classification performance. (a)-(c) Low to Medium SNR; (d)-(f) High SNR.

requirements of 5G NR. This confirms that DyCo-CL occupies an *optimal sweet spot*: it delivers the robustness of heavy models (like ResNet) with the efficiency required for practical deployment.

*Table 4.* Comprehensive Comparison of Deployment Metrics. (Storage is estimated based on FP32 weights).

| Model | Params (M) | FLOPs (M) | Storage (MB) | Latency (ms) | Throughput (samples/s) |
|---|---|---|---|---|---|
| APFS | 1.09 | 50.27 | 4.4 | 42.5 | 23.49 |
| CMSSAN | 0.123 | 2.33 | 0.5 | 0.007 | 140335 |
| EET-MoCo | 1.005 | 11.340 | 4.1 | 1.37 | 729 |
| ResNet50-MoCo | 23.520 | 101.900 | 94.1 | 0.06 | 17280 |
| SSCL-AMC | 1.515 | 36.934 | 6.1 | 2.01 | 498 |
| **DyCo-CL** | **1.443** | **14.46** | **5.8** | **0.60** | **1672** |

## H. Additional Experiments on RML2018.01A

**Confusion Matrix Analysis across SNRs.** As shown in Fig. 10, our model achieves near-perfect separation for non-QAM signals (e.g., PSK, FSK) at 6dB, with remaining errors concentrated within the *QAM family* due to topological inclusion. Despite this inherent ambiguity, DyCo-CL maintains a robust 7.54% lead over the strongest baseline.

