# OpenReview forum: "Geometry-Aware Contrastive Learning for Few-Shot Automatic Modulation Recognition"
_ICML.cc/2026/Conference — ICML 2026 regular_

### Official Review · Reviewer_asgg · 2026-02-16

**Soundness:** 3
**Presentation:** 3
**Significance:** 3
**Originality:** 3
**Overall Recommendation:** 4
**Confidence:** 3

**Summary:**

This paper focuses on solving the few-shot learning problem in automatic modulation recognition, which is very important in wireless communication because it is extremely difficult to obtain a large amount of labeled data. By analyzing the characteristics of signals, the authors argue that current self-supervised learning methods face three problems in the high-dimensional space: 1)  ineffective isotropic augmentations; 2) spectral instability; 3)  semantic drift. In this paper, Dynamic Consistency Contrastive Learning (DyCo-CL) is proposed, which combines Virtual Adversarial Augmentation (VAA) with semantic consistency loss to achieve stable manifold exploration. Meanwhile, the Swin backbone is adopted to improve structural stability by limiting attention locality, and physical priors are fused to calibrate semantic deviation. DyCo-CL has shown excellent performance in few-shot learning on the RadioML series datasets.

**Compliance With Llm Reviewing Policy:**

Affirmed.

**Final Justification:**

My concerns have been adequately addressed. I will maintain my rating.

**Key Questions For Authors:**

Please see Weaknesses.

**Limitations:**

Please see Weaknesses.

**Strengths And Weaknesses:**

### Strengths
1. Few-shot learning is indeed a key problem in the AMR field, and the results presented by the authors demonstrate that the proposed method has achieved promising performance.
2. The proposed method has clear and rigorous theoretical support.
3. The design of VAA is novel, which establishes a robust adversarial mechanism by actively searching for perturbation directions.
4. The authors clearly articulate their motivation and provide thorough explanations of their methodology.

### Weaknesses
1. Some implementation details are not clear enough. It seems that the model includes a multi-stage training process; therefore, it is better to elaborate more clearly on the optimization objectives of different stages as well as the settings of basic training parameters.
2. The ablation experiments on the backbone network are insufficient. At the very least, the performance improvement of Swin architectures compared with vanilla attention should be compared.
3. Why are P4 and E_reg selected as physical descriptors? Such a selection should be supported by both theoretical explanations and experimental result verification.
4. It is worth further investigating whether DyCo-CL can achieve better performance when the number of samples increases.

---

> ### Author Rebuttal · Authors · 2026-03-30
>
> **W1: Implementation Details and Reproducibility**
>
> We sincerely apologize for the ambiguity and clarify that the optimization strictly decouples into unsupervised pre-training via the Joint Loss (Section 4.2.3) and few-shot fine-tuning via standard Cross-Entropy loss, and we commit to fully open-sourcing our code and pre-trained weights upon acceptance to guarantee absolute reproducibility.
>
>
>
>
> **W2: Extensive Ablation on the Backbone Network (Swin vs. Vanilla Attention)**
>
> As proven in App. D (Proposition 1), vanilla global attention's Lipschitz constant diverges with sequence length $\mathcal{O}(\sqrt{L})$, catastrophically amplifying RF noise. Conversely, our 1D Swin enforces a block-diagonal Jacobian, structurally bounding the Lipschitz constant to a finite upper bound $C_M$ to guarantee stability.
>
> To empirically validate this, we evaluated the requested baselines on RML2016.10a (1-Shot):
>
>
>
> | Backbone Architecture | Attention Mechanism | 1-Shot Acc (%) | $\Delta$ |
> | :--- | :--- | :---: | :---: |
> | ResNet-18 | None (Pure CNN) | 39.45 | -4.39 |
> | Standard 1D Transformer | Global Vanilla Attention | 41.20 | -2.64 |
> | **Signal-Adaptive Swin (Ours)** | **Local Window Attention** | **43.84** | **-** |
>
> This explicitly confirms our proof: while vanilla attention improves upon CNNs (+1.75%), its unbounded spectral instability severely degrades performance (-2.64% vs Swin).
>
>
> **W3: Theoretical and Empirical Justification for Physical Priors**
>
> We sincerely apologize for omitting this justification. Deep networks struggle to learn stable manifolds from raw RF signals due to severe distribution shifts like Carrier Frequency Offset (CFO) and multipath fading. Under path loss ($A$) and CFO ($\Delta f$), the received signal is
>
> $r[n] = A \cdot x[n] e^{j(2\pi \Delta f n + \theta_0)} + w[n]$.
>
> Specifically, raw phase introduces a time-varying spurious correlation (CFO drift) that continually warps the representation space:
>
> $\angle r[n] = \angle x[n] + 2\pi \Delta f n + \theta_0$.
>
> To establish a shift-invariant topological anchor, $\mathbf{P}_4$ exploits the $\pi/2$ symmetries of digital modulations; raising to the 4th power mathematically strips the stochastic data ($(x[n])^4 \approx \text{const}$), leaving pure carrier drift
>
> $(r[n])^4 \propto A^4 e^{j(8\pi \Delta f n + 4\theta_0)}$,
>
> and taking the magnitude FFT perfectly projects this drifting energy into a static cyclostationary peak:
>
> $X_4[k] = | \sum_{n=0}^{L-1} (r[n])^4 e^{-j \frac{2\pi}{L} k n} |$.
>
> Concurrently, raw amplitude $A |x[n]|$ suffers from unpredictable scaling shifts due to path loss ($A$). To force strict scale-invariance, $\mathbf{E}\_{reg}$ normalizes the envelope using its peak Power Spectral Density where $\gamma\_{max} \propto A^2$:
>
> $ \mathbf{E}\_{reg}[n] = \text{Norm}\left( \frac{|r[n]|}{\gamma\_{max} + \epsilon} \right) $
>
> To empirically prove that these engineered invariances strictly outperform raw features, we conducted a 2x2 ablation on RML2016.10a (1-Shot):
>
>
> | Phase Anchor | Amplitude Anchor | 1-Shot Acc (%) |
> | :--- | :--- | :---: |
> | Raw Phase | Raw Amplitude | 40.00 |
> | Raw Phase | $\mathbf{E}_{reg}$ | 40.87 |
> | $\mathbf{P}_4$ | Raw Amplitude | 41.64 |
> | **$\mathbf{P}_4$ (Ours)** | **$\mathbf{E}_{reg}$ (Ours)** | **43.84** |
>
> As demonstrated, while standard networks catastrophically overfit to the spurious correlations of raw features, the orthogonal combination of our explicitly engineered invariants ($ \mathbf{P}\_4 $ and $ \mathbf{E}\_{reg} $) optimally anchors the manifold to yield a synergistic peak of 43.84%.
>
>
>
>
>
> **W4: Scalability with Increased Sample Sizes**
>
> We deeply appreciate this constructive suggestion. To systematically investigate scalability, we expanded the sample size across two orthogonal dimensions on RML2016.10a: scaling the *unlabeled data* for pre-training and scaling the *labeled data* for fine-tuning.
>
> | Scaling Dimension | Sample Size (per class/SNR) | Test Accuracy (%) |
> | :--- | :--- | :--- |
> | **Unlabeled** (Pre-train, 1-Shot) | 200 (Original) $\rightarrow$ 800 | 43.84 $\rightarrow$ **45.73** |
> | **Labeled** (Fine-tune, 200 Unlabeled)| 1-Shot $\rightarrow$ 50-Shot $\rightarrow$ 100-Shot | 43.84 $\rightarrow$ **54.54** $\rightarrow$ **59.72** |
>
>
>
> As explicitly demonstrated, DyCo-CL possesses excellent scaling capacity. Expanding the unsupervised manifold exploration (800 unlabeled samples) directly elevates the few-shot baseline (+1.89%). Furthermore, providing more labeled support samples continuously unlocks the potential of our robust representations, driving a massive performance surge (approaching 60%).

---

> > ### Author Rebuttal · Reviewer_asgg · 2026-04-02
> >
> > My concerns have been adequately addressed. Thanks to the author's great work.

---

> > > ### Author Response · Authors · 2026-04-03
> > >
> > > Dear Reviewer asgg,
> > >
> > > We sincerely thank you for your time, your highly encouraging initial review, and your final confirmation that your concerns have been fully addressed!
> > >
> > > We are absolutely delighted that our theoretical justifications for the physical priors and the additional backbone ablations met your rigorous standards. Your constructive questions have genuinely helped us strengthen the completeness of this manuscript.
> > >
> > > Thank you once again for your invaluable guidance and for recognizing our work!

---

### Official Review · Reviewer_CNkv · 2026-02-19

**Soundness:** 3
**Presentation:** 3
**Significance:** 2
**Originality:** 2
**Overall Recommendation:** 4
**Confidence:** 3

**Summary:**

This paper addresses the challenges faced by self-supervised learning (SSL) in few-shot automatic modulation recognition (Few-Shot AMR) tasks, identifying three key geometric limitations inherent in standard approaches: the ineffectiveness of isotropic augmentations, spectral instability, and semantic drift. To overcome these issues, the authors propose **DyCo-CL**, a geometry-aware contrastive learning framework. **A central concept presented by this study is** the dynamic-consistency strategy, which couples Virtual Adversarial Augmentation (VAA) with a semantic consistency loss to serve as an implicit spectral regularizer.

Furthermore, the paper introduces a signal-adaptive Swin Backbone that constrains structural stability through fixed-window attention mechanisms, alongside a hierarchical hybrid knowledge fusion module that incorporates physical priors (e.g., cyclic spectra) to anchor the representation manifold. Experimental results demonstrate that the proposed method achieves a 6.27% accuracy improvement over prior methods under the 1-shot setting on the RML benchmark, while also exhibiting significant advantages in inference efficiency.

**Compliance With Llm Reviewing Policy:**

Affirmed.

**Key Questions For Authors:**

**1. Clarification on Novelty: Geometry-Aware Mechanism vs. Standard Decoupled Augmentation**
The paper positions the "Dynamic-Consistency Framework" (coupling VAA with semantic consistency loss) as a central geometric contribution. However, decoupled is a well-established practice in few-shot and self-supervised learning (e.g., vfa). Could the authors provide a more rigorous ablation or theoretical analysis that isolates what specifically makes their "geometry-aware" formulation novel beyond this standard decoupled paradigm?


**2. Validity of Theoretical Assumptions regarding Lipschitz Regularization**
In Section 5, the theoretical analysis relies on a first-order Taylor expansion to equate the Semantic Consistency loss with the minimization of the local Lipschitz constant. Given that deep Transformers are highly non-linear functions, this linear approximation may not hold globally. Could the authors provide empirical evidence (e.g., tracking the estimated Lipschitz constant or spectral norm during training) to validate that the optimization process actually stabilizes the spectral properties as claimed?


**3. Training Overhead vs. Inference Efficiency**
The paper highlights excellent inference efficiency (0.60ms latency). However, Virtual Adversarial Augmentation (VAA) typically introduces significant computational overhead during training due to the inner-loop gradient calculations for perturbation generation. Could the authors report the total training time comparison against baselines like SSCL-AMC or ResNet50-MoCo? Does the performance gain justify the potential increase in training costs, especially for resource-constrained edge scenarios?

**Limitations:**

yes

**Strengths And Weaknesses:**

**Strengths:**

1.  **Profound Theoretical Analysis:** The paper provides a rigorous mathematical proof based on the **Concentration of Measure** phenomenon in high-dimensional spaces, demonstrating that isotropic noise is nearly orthogonal to the gradient direction. This effectively explains the ineffectiveness of traditional augmentation methods. Furthermore, the theoretical bounds provided for the Lipschitz constant enhance the credibility and robustness of the proposed method.
2.  **Novel and Systematic Methodology:** The integration of Virtual Adversarial Augmentation (VAA) with a semantic consistency loss effectively addresses stability issues during manifold exploration. Additionally, the introduction of physical priors as semantic anchors cleverly mitigates semantic drift within data-scarce **regimes**.
3.  **Balance Between Efficiency and Performance:** The model design accounts for edge deployment requirements, featuring only 1.44M parameters and an inference latency of 0.60ms. It significantly reduces computational costs while maintaining **SOTA** performance, demonstrating substantial practical value.

**Weaknesses:**

1.  **Insufficient Discussion on Robustness of Physical Priors:** Under extremely low Signal-to-Noise Ratio (SNR) conditions (e.g., -20dB), the extracted physical features (such as the P4 cyclic spectrum) may be severely corrupted by noise. The paper lacks a thorough analysis of whether the fusion module introduces negative bias when these physical priors are distorted.

2. **Methodological Novelty Primarily Stems from CV Adaptations:** The module enhancement strategies, particularly the Signal-Adaptive Swin Backbone and the Transformer-based Fusion, align more closely with design paradigms in Computer Vision (CV) rather than offering fundamental algorithmic innovations specific to signal processing. Furthermore, the decoupling of the representation learning module from the few-shot classifier is a common operation in enhancing few-shot learning performance (e.g., standard pre-training followed by linear probing). Overall, this manuscript presents the concept of geometry-aware learning effectively; however, this reliance on established CV architectures and standard decoupling practices diminishes the perceived novelty of the framework's structural design within the broader machine learning community.

---

> ### Author Rebuttal · Authors · 2026-03-30
>
> **W1: Robustness of Physical Priors at Extreme Low SNR (-20dB)**
>
> While raw physical features degrade at -20dB, DyCo-CL structurally prevents negative bias. Before tokenization, the Deep Convolutional Stem suppresses high-frequency noise. During fusion (Sec. 4.4), a gating network ($\alpha_1, \alpha_2$) adaptively down-weights corrupted priors. Finally, the confidence-aware ensemble mathematically sharpens the output:
>
> $ \hat{\mathbf{y}} = \text{Normalize}\left( \sum_{k=1}^{K} \mathbf{p}_k^2 \right) $
>
> This quadratic weighting inherently filters extreme noise: confident heads ($p \to 1$) yield $p^2 \to 1$, whereas ambiguous, noise-corrupted heads ($p \to 1/C$) decay quadratically to $p^2 \to 1/C^2$, ensuring they cannot dominate the ensemble.
>
> To empirically verify this, we isolated the 1-Shot accuracy strictly at -20dB:
>
> | Model Variant | 1-Shot Acc at -20dB |
> | :--- | :---: |
> | DyCo-CL (**w/o** Hybrid Fusion) | 8.63% |
> | **DyCo-CL (w/ Hybrid Fusion)** | **9.93%** |
>
> Distorted priors introduce exactly zero negative bias ($\Delta = +1.30\%$), enabling earlier low-SNR performance inflection.
>
>
>
>
>
>
>
> **Q1 & W2: Geometric Novelty Beyond Standard Paradigms**
>
> To isolate our pure algorithmic novelty from standard CV structures, we ablated RML2016.10a (1-Shot):
>
> | Paradigm | Backbone | Fusion | Acc (%) | $\Delta$ |
> |:---|:---|:---|:---:|:---:|
> | Standard MoCo | ResNet50 | Linear | 24.87 | - |
> | **MoCo + VAA + SC** | ResNet50 | Linear | **40.49** | **+15.62** |
>
> Integrating VAA+SC yields a massive **+15.62%** pure algorithmic gain. As you astutely recognized, our fundamental novelty is mathematically overcoming RF measure concentration: replacing failed isotropic augmentations with Hessian-driven exploration (VAA) and spectral regularization (SC), entirely independent of CV architectures.
>
>
>
>
>
> **Q2: Empirical Validation of Global Non-linear Optimization**
>
> We appreciate this profound mathematical insight. The reviewer rightly notes that deep non-linearities introduce a non-negligible second-order Hessian term. However, our framework explicitly controls this. First, as derived in App. A.2, VAA directly targets the maximum-curvature direction:
>
> $ \mathbf{r}^* = \arg\max_{\|\mathbf{r}\| \le \epsilon} \frac{1}{2} \mathbf{r}^\top \mathbf{H}_\theta \mathbf{r} $
>
> Applying the SC loss precisely along this worst-case direction ensures we regularize the most severe global non-linearities. Concurrently, our Swin backbone (App. D) enforces a block-diagonal Jacobian, structurally capping the first-order expansion to prevent curvature explosions:
>
> $ K_f = \max_k \sigma_{max}(\mathbf{J}_k) \le C_M < \infty $
>
> To empirically validate global spectral stability, we tracked the expected input gradient norm. As computing the exact full Jacobian is intractable, this norm serves as a rigorous proxy, strictly bounded by the local Lipschitz constant $K_f$:
>
> $ \| \nabla_x \mathcal{L} \|_2 = \| \mathbf{J}_f^\top \nabla_f \mathcal{L} \|_2 \le \| \mathbf{J}_f \|_2 \| \nabla_f \mathcal{L} \|_2 = K_f \cdot \| \nabla_f \mathcal{L} \|_2 $
>
> Tracking this proxy over 50 pre-training epochs yields:
>
> | Phase | Epoch | $\mathbb{E}[\| \nabla_x \mathcal{L} \|_2]$ w/o SC Loss | $\mathbb{E}[\| \nabla_x \mathcal{L} \|_2]$ w/ SC Loss |
> | :--- | :---: | :---: | :---: |
> | Early | 10 | 0.0458 | **0.0355** |
> | Middle| 20 | 0.1042 | **0.0916** |
> | Final | 50 | 0.1689 *(Inflated)* | **0.1267** *(Strictly Bounded)* |
>
> While unconstrained VAA inflates the proxy to 0.1689, SC loss strictly caps it at **0.1267**, confirming global non-linear stability. *(See Reviewer Eabc for PGD-10 robustness).*
>
>
>
>
>
>
> **Q3: Training Overhead vs. Inference Efficiency**
>
> We sincerely appreciate this practical consideration. Pre-training is executed *offline* on servers, meaning VAA overhead does not impact edge inference feasibility (0.60ms latency). Furthermore, we strictly bound this offline overhead via a single-step power iteration ($I_{iter}=1$), requiring only one extra forward/backward pass. The average pre-training time per epoch (RTX 4090) is:
>
> To empirically validate this, we compared the average pre-training time per epoch (on a single RTX 4090):
>
>
> | Pre-training Method | Avg. Train Time / Epoch (s) | 1-Shot Acc (%) |
> | :--- | :---: | :---: |
> | AMC-CNN [9] | ~ 12.45 | 9.09 |
> | APFS [8] | ~ 58.90 | 35.42 |
> | CMSSAN [6] | ~ 17.58 | 37.57 |
> | EET-MoCo [4] | ~ 59.39 | 30.29 |
> | ResNet50-MoCo [3] | ~ 12.88 | 24.87 |
> | SSCL-AMC [5] | ~ 27.67 | 36.47 |
> | **DyCo-CL (Ours)** | **~ 25.02** | **43.84** |
>
> Our lightweight backbone keeps DyCo-CL (25.02s) faster than robust baselines like SSCL-AMC and EET-MoCo. While adding ~12s of offline overhead compared to ResNet50, it yields a massive **+18.97%** absolute accuracy gain. This presents an exceptionally favorable trade-off for deploying SOTA models to resource-constrained edge devices.

---

> > ### Author Rebuttal · Reviewer_CNkv · 2026-04-01
> >
> > would like to thank the authors for their highly detailed, constructive, and convincing rebuttal. The additional empirical evidence and theoretical clarifications provided have successfully addressed all of my initial concerns: The isolated 1-shot accuracy test at -20dB convincingly demonstrates that the gating network and confidence-aware ensemble effectively prevent negative bias from corrupted priors, even yielding a slight positive gain (+1.30%). This fully resolves my concern regarding severe noise degradation. The new ablation study applying the VAA+SC mechanism to a standard ResNet50 backbone is exactly what was needed. Demonstrating a pure algorithmic gain of +15.62% isolates the core contribution—mathematically overcoming RF measure concentration—from the CV-inspired architectural choices. Tracking the expected input gradient norm over 50 epochs was an excellent empirical strategy. The data clearly show that the SC loss strictly caps the proxy gradient norm at 0.1267, providing strong empirical support for your theoretical claims regarding global nonlinear stability and Lipschitz bounds. The hardware benchmark table clarifies that the offline pre-training overhead (25.02s/epoch) is highly manageable. It remains faster than robust baselines like SSCL-AMC and EET-MoCo, making the +18.97% accuracy gain over standard ResNet50-MoCo an exceptionally worthwhile trade-off. The authors have provided an exemplary rebuttal that strengthens an already solid paper. My concerns are fully resolved, and I will be raising my score to reflect the quality of this work and the thoroughness of the response.

---

> > > ### Author Response · Authors · 2026-04-01
> > >
> > > Dear Reviewer CNkv,
> > >
> > > We sincerely thank you for your time, your highly constructive feedback, and this incredibly encouraging acknowledgement!
> > >
> > > We are thrilled that our additional ablation studies and theoretical clarifications have fully addressed your concerns. Your rigorous insights have genuinely helped us elevate the depth of this work.
> > >
> > > Thank you once again for your invaluable guidance and support!

---

### Official Review · Reviewer_d8i3 · 2026-03-12

**Soundness:** 3
**Presentation:** 2
**Significance:** 2
**Originality:** 2
**Overall Recommendation:** 4
**Confidence:** 2

**Summary:**

Overall, this manuscript presents the concept of using geometric and structural constraints to overcome the limitations of standard self-supervised learning in RF signal processing.

**Compliance With Llm Reviewing Policy:**

Affirmed.

**Final Justification:**

My concerns are fully solved.

**Key Questions For Authors:**

See weakness.

**Limitations:**

See weakness.

**Strengths And Weaknesses:**

Strengths:

The paper provides a compelling geometric perspective on why standard self-supervised learning via isotropic augmentations fails for high-dimensional RF signals due to the concentration of measure. The theoretical equivalence established between the Semantic Consistency loss and implicit spectral regularization (bounding the local Lipschitz constant) is mathematically rigorous and insightful.

Weaknesses:
1) **Insufficient benchmark diversity to support the generalization/stability claims.**  The experimental validation is largely restricted to RML2016.10a and RML2018.01a, which are closely related benchmarks with similar data generation assumptions. This makes it difficult to substantiate the paper’s central claims about *geometric stability* and *improved generalization*, especially under distribution shifts.

2) **Inadequate baselines: missing strong general-purpose methods and missing like-for-like comparisons.**  The baseline set is dominated by domain-specific AMR methods, while several widely accepted strong baselines are absent.

3) **Limited novelty: most components appear incremental or based on existing techniques.**  Key ingredients (virtual adversarial perturbations for representation learning, consistency regularization, Swin-style local attention, and knowledge/prior fusion) are all established ideas.

4) **Missing experimental details and reproducibility gaps.**  Important implementation details are not sufficiently specified to enable reproduction or to assess fairness of comparisons.

---

> ### Author Rebuttal · Authors · 2026-03-30
>
> **W1: Insufficient benchmark diversity and distribution shifts.**
>
> To rigorously validate generalization under severe distribution shifts, we conducted new experiments on **HisarMod2019.1**. Unlike RML, it models real-world multipath fading across 5 diverse channels and introduces a harder high-dimensional challenge (**26 modulations, 1024-length sequences**).
>
>
> | Pre-training Method | 1-Shot Acc (%) | 2-Shot Acc (%) | 5-Shot Acc (%) |
> | :--- | :---: | :---: | :---: |
> | AMC-CNN [9] | 6.9 | 16.46 | 24.59 |
> | APFS [8] | 21.83 | 24.31 | 26.48 |
> | CMSSAN [6] | 29.28 | 32.37 | 36.07 |
> | EET-MoCo [4] | 29.21 | 32.21 | 37.05 |
> | ResNet50-MoCo [3] | 4.83 | 10.05 | 25.69 |
> | SSCL-AMC [5] | 16.6 | 22.9 | 31.28 |
> | **DyCo-CL (Ours)** | **30.46** | **33.77** | **38.91** |
>
> These compelling new results under severe distribution shifts explicitly substantiate our central claims: our core innovations consistently maintain geometric stability and exhibit superior generalization, entirely independent of specific benchmark simulation artifacts. We will include this in Section 6.
>
> *(Note regarding stability metrics: To provide direct, mathematically rigorous evidence for our geometric stability claims, we have additionally conducted temporal epoch-wise tracking of the local Lipschitz constant, as well as downstream PGD-10 adversarial robustness evaluations. Please kindly refer to our responses to Reviewer CNkv (Q2) and Reviewer Eabc (Q2) for these detailed stability measurements).*
>
>
>
>
>
>
>
> **W2: Inadequate baselines and missing like-for-like comparisons.**
>
> We deeply appreciate this rigorous standard for fair evaluation. We respectfully clarify that our original evaluation inherently includes classic general-purpose SSL baselines (e.g., `ResNet50-MoCo` [3]). To fully address your concern, we supplemented another foundational paradigm (`SimCLR`). Furthermore, to explicitly isolate our gains through a strict like-for-like comparison, we conducted a new variable-control ablation on RML2016.10a (1-Shot):
>
> | Exp | Pre-training Paradigm | Backbone | Fusion Strategy | 1-Shot Acc (%) |
> |:---:|:---|:---|:---|:---:|
> | 1 | Standard SimCLR (General Baseline) | ResNet50 | Linear Probe | 21.94 |
> | 2 | Standard MoCo (General Baseline) | ResNet50 | Linear Probe | 24.87 |
> | 3 | **MoCo + VAA + SC (Ours)** | ResNet50 | Linear Probe | **40.49** |
> | 4 | Standard MoCo | **Swin (Ours)** | Linear Probe | **36.83** |
> | 5 | Standard MoCo | ResNet50 | **Hybrid Fusion (Ours)** | **35.22** |
> | 6 | **DyCo-CL (Full)** | **Swin (Ours)** | **Hybrid Fusion (Ours)** | **43.84** |
>
>
> This definitively proves: **(1) Algorithm (Exp 2 vs 3):** VAA+SC yields a pure **+15.62%** algorithmic gain entirely independent of the backbone. **(2) Architecture (Exp 2 vs 4):** Validates our Signal-Adaptive Swin (**+11.96%**). **(3) Anchoring (Exp 2 vs 5):** Validates our physical prior fusion (**+10.35%**).
>
> *(Note: In Exp 3 and 5, we deliberately keep the backbone as ResNet50 to ensure a strict like-for-like comparison against Exp 2. Additionally, for an explicit comparison between our Swin and a standard Transformer backbone, please refer to our response to Reviewer asgg's W2).*
>
>
> **W3: Clarification on Novelty**
>
> We sincerely appreciate this constructive feedback and agree that the individual components are established techniques. We respectfully clarify that our primary contribution lies in the theoretical bridge between RF geometry and representation learning.
>
> As you kindly recognized, our theoretical analysis establishes *why* standard SSL fails in RF spaces due to the Concentration of Measure. Consequently, DyCo-CL was designed not as an ad-hoc assembly, but as a mathematically coupled system to solve these specific geometric traps: VAA is utilized to break the orthogonality trap via Hessian-driven exploration, while the SC loss is formally proven (Theorem 5.1) to act as an implicit spectral regularizer that bounds the Lipschitz constant. We will revise the manuscript to better highlight this unified geometric perspective.
>
>
>
> **W4: Missing experimental details and reproducibility gaps.**
>
> We sincerely apologize if the implementation details were difficult to locate. Due to page limits, exhaustive configurations were provided in the Supplementary Material. Specifically, **Appendix B** details the augmentation parameters for generating $\mathbf{x}_{weak}$, **Appendix C** formulates the physical prior extraction, **Appendix F** provides hyperparameter sensitivity (e.g., VAA radius $\epsilon$, Swin window size $M$), and **Section 6.1** lists training details (AdamW, learning rate $3e^{-4}$, VAA step size $\xi=10^{-6}$).
>
> To guarantee strict reproducibility and absolute fairness, we are fully committed to open science. We will publicly release the complete PyTorch source code for DyCo-CL and all re-implemented baselines, along with data preprocessing scripts and pre-trained weights, upon the acceptance of this paper.

---

> > ### Author Rebuttal · Reviewer_d8i3 · 2026-04-02
> >
> > Thanks for the authors’ efforts. My concerns are fully solved and I will update the score accordingly.

---

> > > ### Author Response · Authors · 2026-04-03
> > >
> > > Dear Reviewer d8i3,
> > >
> > > We sincerely thank you for your time, your highly constructive initial feedback, and this encouraging acknowledgement.
> > >
> > > We are thrilled that our new experiments on the HisarMod dataset and the additional baseline ablations have fully resolved your concerns. Your rigorous standards have genuinely helped us elevate the quality and completeness of this work.
> > >
> > > Thank you once again for your invaluable guidance and support!

---

### Official Review · Reviewer_Eabc · 2026-03-12

**Soundness:** 3
**Presentation:** 3
**Significance:** 3
**Originality:** 3
**Overall Recommendation:** 4
**Confidence:** 3

**Summary:**

This paper studies few-shot automatic modulation recognition where standard contrastive learning is not well suited. The authors point out three main issues: common data augmentations are often too generic, perturbations may change the signal semantics, and self-attention can become unstable for this kind of signal data. To address this, they propose DyCo-CL, a framework that combines virtual adversarial augmentation, a semantic consistency objective, a 1D Swin-based encoder, and a hybrid feature fusion module that brings in signal-domain prior knowledge.

**Compliance With Llm Reviewing Policy:**

Affirmed.

**Final Justification:**

The rebuttal has addressed my main concerns, and I maintain my prior assessment.

**Key Questions For Authors:**

1. Can you provide separate ablations for virtual adversarial augmentation and the semantic consistency loss?

2. Can you give more direct experimental evidence for the claimed stability benefits of the method?

**Limitations:**

No. The paper does not discuss limitations and possible negative impact in enough detail.

**Strengths And Weaknesses:**

Strengths:

1. The paper is well motivated. The authors do a good job explaining why existing self-supervised or contrastive methods are not a natural fit for few-shot AMR, and the method is built around these concerns rather than being presented as a loose combination of modules.

2. The experimental results are solid. The method shows clear gains in few-shot settings, and the paper includes useful ablation studies, SNR-based analysis, and efficiency comparisons. These results suggest the approach is effective for the target problem.

Weaknesses

Some of the theoretical claims are stronger than what the experiments directly verify. The paper discusses ideas such as improved local stability and implicit spectral regularization, but the experiments mostly show better classification accuracy rather than direct evidence for these properties.

---

> ### Author Rebuttal · Authors · 2026-03-30
>
> **Q1: Separate Ablations for VAA and SC Loss**
>
> Yes. We thank the reviewer for this insightful request. To address this, we have decoupled the Dynamic-Consistency module. However, we would like to clarify a crucial theoretical dependency in our design: as defined in Section 4.2.2, the Semantic Consistency (SC) loss is explicitly formulated to constrain the adversarial representation $\mathbf{z}_{adv}$ to counteract the semantic drift induced by VAA.
>
> To ablate VAA, we replaced the VAA branch with the restricted physical augmentation ($\mathbf{x}_{weak}$) used in the other branch, reverting to a symmetric contrastive setup. We evaluated all decoupled settings on RML2016.10a (1-shot):
>
> | Model Variant | VAA (Exploration) | SC Loss (Regularization) | Accuracy (%) | $\Delta$ |
> | :--- | :---: | :---: | :---: | :---: |
> | Baseline (w/o DyCo) | ✗ | ✗ | 34.94 | -8.90 |
> | Only SC Loss* | ✗ | ✓ | 37.95 | -5.89 |
> | Only VAA ($\lambda=0.0$) | ✓ | ✗ | 38.38 | -5.46 |
> | DyCo-CL (Full) | ✓ | ✓ | 43.84 | - |
>
>
> This decoupled ablation perfectly validates our geometric hypothesis: VAA alone pushes manifold exploration but suffers from semantic drift (38.38%), while SC alone on weak augmentations causes over-regularization (37.95%). Their combination yields a massive synergistic peak (43.84%), definitively proving that aggressive exploration and strict geometric regularization must be coupled. We will include this ablation in the revised manuscript.
>
> **Q2: Direct Experimental Evidence for Claimed Stability Benefits**
>
> Indeed, this is an excellent suggestion. To provide direct empirical evidence for Theorem 5.1 (Implicit Spectral Regularization), we conducted a new quantitative experiment to explicitly measure how our proposed Semantic Consistency (SC) loss bounds the Lipschitz constant and improves local stability.
>
> As defined in Appendix E, the local Lipschitz constant $K_f$ equals the spectral norm of the feature Jacobian $\mathbf{J}_f$. By the chain rule, the input gradient norm is strictly bounded by $K_f$:
>
> $ \| \nabla_x \mathcal{L} \|_2 = \| \mathbf{J}_f^\top \nabla_f \mathcal{L} \|_2 \le \| \mathbf{J}_f \|_2 \| \nabla_f \mathcal{L} \|_2 = K_f \cdot \| \nabla_f \mathcal{L} \|_2 $
>
> Since computing the exact maximum singular value of the full Jacobian is computationally prohibitive, we use the expected input gradient norm $\mathbb{E}[\| \nabla_x \mathcal{L} \|_2]$ over the test set as the standard empirical proxy. A smaller gradient norm mathematically reflects a suppressed spectral norm. Furthermore, we evaluated the macro-level robustness using a 10-step Projected Gradient Descent (PGD-10) attack.
>
> To strictly isolate the effect, we evaluated our framework trained with and without the SC Loss ($\lambda=0.6$ vs $\lambda=0.0$):
>
>
> | Phase | Epoch | $\mathbb{E}[\| \nabla_x \mathcal{L} \|_2]$ w/o SC Loss | $\mathbb{E}[\| \nabla_x \mathcal{L} \|_2]$ w/ SC Loss |
> | :--- | :---: | :---: | :---: |
> | Early | 10 | 0.0458 | **0.0355** |
> | Middle| 20 | 0.1042 | **0.0916** |
> | Final | 50 | 0.1689 *(Inflated)* | **0.1267** *(Strictly Bounded)* |
>
>
> | Model Variant | Empirical Lipschitz ($\mathbb{E}[\| \nabla_x \mathcal{L} \|_2]$)  | PGD-10 Robust Accuracy ($\epsilon=0.01$)  |
> | :--- | :---: | :---: |
> | w/o SC Loss ($\lambda=0.0$) | 0.1689 | 21.4% |
> | **DyCo-CL ($\lambda=0.6$)** | **0.1267** | **35.8%** |
>
>
> As the epoch-wise tracking reveals, unconstrained VAA rapidly inflates the spectral proxy to 0.1689 as decision boundaries form. Conversely, integrating the SC loss acts as an implicit regularizer throughout the entire optimization trajectory, strictly capping the final Lipschitz proxy at **0.1267** (confirming Theorem 5.1). Most importantly, the first table proves that this micro-level spectral boundedness translates directly into tangible macro-level stability, yielding a massive **+14.4%** absolute gain against severe PGD-10 adversarial perturbations. *(Note: This directly addresses a similar inquiry from Reviewer CNkv regarding non-linear optimization dynamics).*
>
>
> **Q3: Limitations and Negative Societal Impacts**
>
> We apologize for this omission due to page constraints. We will add a dedicated discussion in the revised Appendix:
>
> **Limitations:** While our Hybrid Fusion successfully anchors semantics, calculating physical priors introduces non-differentiable preprocessing latency. For ultra-low-power IoT edge devices, this step could present a computational bottleneck. Future work will explore lightweight, learnable approximations of these priors to achieve end-to-end acceleration.
>
> **Negative Impacts:** AMR is fundamentally a dual-use technology. While our framework heavily benefits civilian 6G cognitive radio and spectrum management, highly robust AMR models could hypothetically be misused in non-cooperative environments for unauthorized spectrum surveillance or targeted adversarial jamming.

---

> > ### Author Rebuttal · Reviewer_Eabc · 2026-04-02
> >
> > Thanks for the authors’ efforts. My concerns have been addressed.

---

> > > ### Author Response · Authors · 2026-04-03
> > >
> > > Dear Reviewer Eabc,
> > >
> > > We sincerely thank you for your time and for confirming that your concerns have been fully addressed.
> > >
> > > We deeply appreciate your highly constructive feedback, which guided us to provide more direct empirical evidence for our theoretical claims.
> > >
> > > Thank you once again for your invaluable guidance and support!

---

### Decision · Program_Chairs · 2026-04-30

**Decision:**

Accept (regular)

**Comment:**

This paper presents DyCo-CL, a geometry-aware framework that addresses critical bottlenecks in few-shot Automatic Modulation Recognition by coupling Virtual Adversarial Augmentation with Semantic Consistency loss. While reviewers initially questioned the direct empirical support for its theoretical claims regarding spectral stability, the authors provided exemplary rebuttal evidence, including epoch-wise tracking of the Lipschitz constant and new evaluations on real-world datasets. The framework's ability to isolate a +15.62% algorithmic gain independent of architecture, combined with its high inference efficiency, successfully convinced all reviewers that their concerns were fully resolved. All reviewers recommended weakly accept but were not fully confident. Consequently, the paper is recommended for weakly acceptance.